# Structure-based mechanism of RyR channel operation by calcium and magnesium ions

Alexandra Zahradníková[1]*, Jana Pavelková[1], Miroslav Sabo[2], Sefer Baday[3], Ivan Zahradník[1]*

1 Department of Cellular Cardiology, Institute of Experimental Endocrinology, Biomedical Research Center, Slovak Academy of Sciences, Bratislava, Slovakia, 2 Bioinformatics Laboratory, Biomedical Research Center, Slovak Academy of Sciences, Bratislava, Slovakia, 3 Applied Informatics Department, Informatics Institute, Istanbul Technical University, Istanbul, Türkiye

* alexandra.zahradnikova@savba.sk (AZ); ivan.zahradnik@savba.sk (IZ)

## Abstract

Ryanodine receptors (RyRs) serve for excitation-contraction coupling in skeletal and cardiac muscle cells in a noticeably different way, not fully understood at the molecular level. We addressed the structure of skeletal (RyR1) and cardiac (RyR2) isoforms relevant to gating by $Ca^{2+}$ and $Mg^{2+}$ ions ($M^{2+}$). Bioinformatics analysis of RyR structures ascertained the EF-hand loops as the $M^{2+}$ binding inhibition site and revealed its allosteric coupling to the channel gate. The intra-monomeric inactivation pathway interacts with the $Ca^{2+}$-activation pathway in both RyR isoforms, and the inter-monomeric pathway, stronger in RyR1, couples to the gate through the S23*-loop of the neighbor monomer. These structural findings were implemented in the model of RyR operation based on statistical mechanics and the Monod-Wyman-Changeux theorem. The model, which defines closed, open, and inactivated macrostates allosterically coupled to $M^{2+}$-binding activation and inhibition sites, approximated the open probability data for both RyR1 and RyR2 channels at a broad range of $M^{2+}$ concentrations. The proposed mechanism of RyR operation provides a new interpretation of the structural and functional data of mammalian RyR channels on common grounds. This may provide a new platform for designing pharmacological interventions in the relevant diseases of skeletal and cardiac muscles. The synthetic approach developed in this work may find general use in deciphering mechanisms of ion channel functions.

## Author summary

The relationship between the structure and function of ryanodine receptors is crucial for understanding their role in the contraction of cardiac and skeletal muscle cells. These receptors regulate the release of calcium ions from intracellular stores, integrating numerous activatory and inhibitory signals. Despite the

**Data availability statement:** All relevant data are within the manuscript and its Supporting information files.

**Funding:** This work was supported by Slovak Academy of Sciences (SAS) and the Scientific and Technological Research Council of Turkey (TÜBITAK) within the framework of the Turkish-Slovak Joint Partnership: SAS (https://oms.sav.sk/en/programmes-and-scholarships/joint-research-projects/) grant number JRP/2019/836/RyRinHeart to AZ and TÜBITAK (https://tubitak.gov.tr/en) grant number 119Z578 to SB, and by the projects APVV-21-0443 from the Slovak Research and Development Agency (https://www.apvv.sk/?lang=en) and VEGA 2/0182/21 from the Ministry of Education, Research, Development and Youth of the Slovak Republic and Slovak Academy of Sciences (https://vega.sav.sk/). The funders had no role in study design, data collection and analysis, decision to publish, or preparation of the manuscript.

**Competing interests:** The authors have declared that no competing interests exist.

wealth of data generated by various research methods, synthesizing a unified interpretation has remained difficult. In this study, we employed state-of-the-art computational techniques to analyze the published structures of the cardiac and skeletal isoforms of ryanodine receptors, with a focus on their regulation by calcium and magnesium ions. Our analysis revealed activation and inhibition pathways that connect ion-binding sites to the channel gate, which are largely similar between the isoforms, though with subtle, isoform-specific differences. We then incorporated these structural insights into a thermodynamic model of ryanodine receptor function, which was validated against experimental data from multiple laboratories. The model accounts for the distinct functional properties of the two isoforms, with a common set of parameters, of which only two exhibit isoform-specific values. This work provides a comprehensive framework for understanding the operation of ryanodine receptors in muscle contractility, offering a novel approach to structure-function analysis of complex membrane channels and receptors.

## Introduction

Ryanodine receptor (RyR) plays a central role in the calcium-dependent functions of excitable cells. RyRs reside on the membrane of the endoplasmic/sarcoplasmic reticulum (ER/SR) and upon cell excitation, they control the release of calcium ions from the ER/SR lumen to cytosol. Three RyR isoforms are expressed in mammals: RyR1 in skeletal muscle cells, RyR2 in cardiomyocytes, neurons, and endocrine cells, and RyR3 in many tissues at lower levels [1]. Although RyR isoforms differ substantially in their sequence, function, and role in different tissue types, they show almost identical molecular architecture [2]. All three RyR isoforms form a homo-tetrameric transmembrane channel, which harbors multiple regulatory sites, mostly at their cytosolic side [1]. It is important to note that, with a few exceptions, the regulatory binding sites reside on the monomer, so there are 4 equivalent binding sites of each type per RyR tetramer. Consequently, individuals heterozygous for a congenital mutation in the RyR gene have an equal abundance of mutated and wild-type monomers, which combine at random into tetramers [3,4].

Specific ligand-binding sites have been found for natural activators $Ca^{2+}$, ATP, and xanthines [5–8]; for regulatory molecules like calstabins and calmodulin [5,9,10], the diamide insecticide chlorantraniliprole [11]; and for emerging clinically important stabilizers like Rycals [7,8]. RyR activators ryanodine and imperacalcin have binding sites inside the conduction pathway common to the whole RyR tetramer [5,12]. Many other regulatory sites need to be revealed for regulators such as dantrolene, junctin, triadin, luminal $Ca^{2+}$, calsequestrin, etc. The modulation of RyR activity by phosphorylation, oxidation, nitrosylation, etc., is of importance but only the effects of PKA phosphorylation on RyR2 structure have been studied up to date [8].

The molecular architecture as well as the localization of mutations that compromise RyR function is very similar among isoforms. Most known mutations cause

muscle conditions that may have fatal consequences in the case of cardiac muscle [13,14]. Over two hundred mutations that cause dysregulation of calcium release are localized mainly in four specific regions of the RyR monomer [13,15–17]. These mutations are typically characterized by the increased sensitivity of calcium release to cytosolic and/or luminal calcium, by the decreased sensitivity to inactivation by high cytosolic concentrations of magnesium or calcium, or by increased basal RyR activity [4,14,18–20].

The complex structure of RyR tetramer may adopt a large number of configurations [5,21]. In RyR1, four discrete structural states were discerned under different cytosolic conditions - closed (C), primed (P), open (O), and inactivated (I) [5,22,23]. With increasing concentrations of the activators ($Ca^{2+}$, ATP derivatives, or xanthine derivatives), the configuration of the tetramers shifted from the closed state to the primed and open state [5,21]. At a very high $Ca^{2+}$ or $Mg^{2+}$ concentration, the inactivated state was observed [22,23]. Under similar conditions, analogous C, P, and O states were observed also in RyR2 [6]; unfortunately, the structure of RyR2 in the inactivated state has not been published yet.

Upon transition from the closed state to either primed, open, or inactivated states, the large cytoplasmic shell moves downward and outward [5,8,24], a movement that transpires as a progressively more negative flexion angle [24], and the Central and C-terminal domains rotate against each other in each monomer [22]. The overall structure of the primed and inactivated states resembles more the open than the closed state but the ion conduction pore is closed [5,6,22].

In striated muscle cells, RyR channels cluster at the discrete sites of the sarcoplasmic reticulum attached to the sarcolemma in triads or dyads, where electrical excitation triggers transient calcium release by activation of RyRs. Activation mechanisms differ between skeletal and cardiac muscle cells. *In situ*, RyR1 channels are open by direct coupling to the $Ca_v1.1$ channels, while RyR2 channels are activated by binding calcium ions injected through the $Ca_v1.2$ channels [25]. In isolation, however, the RyR1 and RyR2 channels are activated by $Ca^{2+}$ ions equally well [26]. The activation calcium-binding site was structurally identified in both RyR1 and RyR2 [5,6].

The termination of RyR activity in situ is not fully understood [27–29]. In skeletal muscle cells, calcium-dependent inactivation of RyR1 is generally accepted as the major termination mechanism [27]. In cardiac muscle cells, the termination of RyR activity is less clear since the calcium-dependent RyR2 inactivation is weaker than that of RyR1; nevertheless, divalent ions ($M^{2+}$) are involved in the RyR2 inactivation/inhibition as well [30–34].

In both RyR1 and RyR2 isoforms, $Mg^{2+}$ ions decrease the intensity of calcium release in a concentration-dependent manner. In isolated RyRs, $Mg^{2+}$ ions lead to an increase of the EC50 and a reduction of the maximum of RyR activation by $Ca^{2+}$ [30,35,36]. $Ca^{2+}$ and $Mg^{2+}$ have a similar inhibitory potency on the open probability in both RyR1 and RyR2 [35]. This indicates the presence of a non-specific divalent ion-binding site in the RyR structure. Recently, mathematical modeling of the cardiac calcium release site [30] demonstrated that $Mg^{2+}$ ions could at the same time act as the negative competitor at the calcium activation site and as an inhibitor at the inhibition site. Although binding of $Ca^{2+}$ or $Mg^{2+}$ to an inhibitory binding site has not been observed yet in RyR structures, a consensus is emerging that the EF-hand loops constitute this site [23,37–39].

The activation of RyR by agonists was shown to be accompanied by a conformational change around the $Ca^{2+}$ binding site that leads to a decrease in the free energy of the open state and to a concomitant increase of the $Ca^{2+}$ binding affinity of the activation site. As a result, the occurrence probability of a RyR state/conformation shifts from the closed toward the open [21]. It should be pointed out that the activation mechanism of both RyR1 and RyR2 by calcium is fundamentally identical regarding the $Ca^{2+}$ dependence of their open probability [26]. The amino acid identity and 3D configuration of binding sites for $Ca^{2+}$, ATP, and caffeine/xanthine are equivalent in both RyR isoforms [5,6].

Though RyRs have similar activation mechanisms, they have different inactivation mechanisms. The single-channel activity of RyR1 was inhibited by $Ca^{2+}$ at a much lower concentration than that of RyR2 [26,40,41]. Single-channel experiments provided the same concentration dependence of the RyR inhibition for both $Mg^{2+}$ and $Ca^{2+}$ ions ($M^{2+}$), but a substantially lower degree of inhibition in RyR2 than in RyR1 channels [35]). These characteristics are opposite to the corresponding characteristics of RyR activation. Consequently, the underlying structures and mechanisms are expected to

be principally different, which is puzzling since the equivalent components of RyR isoforms are similar. Moreover, the pore of the RyR tetramer is lined, limited, and gated by the quadruplet of S6 segments; thus, one gating site executes both the activation and the inactivation. Obviously, there is a paradox in RyR channels that equivalent structures execute the activation similarly but the inactivation differently in RyR1 and RyR2.

The regulatory domains involved in both, the activation and inactivation of RyRs (Fig 1) are located in the C-terminal quarter of the RyR. The Central domain participates in the $Ca^{2+}$ binding activation site; the C-terminal domain bears several residues of Ca-, ATP- and caffeine-binding activation sites; the U-motif participates at the ATP- and caffeine-binding sites; the EF-hand region contains the putative Ca-binding pair EF1 and EF2; and the S23 loop bears one residue of the caffeine-binding site and two residues interacting with the EF-hand region of a neighboring monomer [42,43].

Recent studies reporting RyR structure at a high divalent ion concentration provide only indirect support for the molecular mechanism of $Ca^{2+}/Mg^{2+}$-dependent inactivation. Wei et al. [44] and Nayak et al. [23] observed a change in the conformation of the RyR1 EF-hands in the presence of 100 µM $Ca^{2+}$ and 10 mM $Mg^{2+}$, respectively, compared to low-calcium or

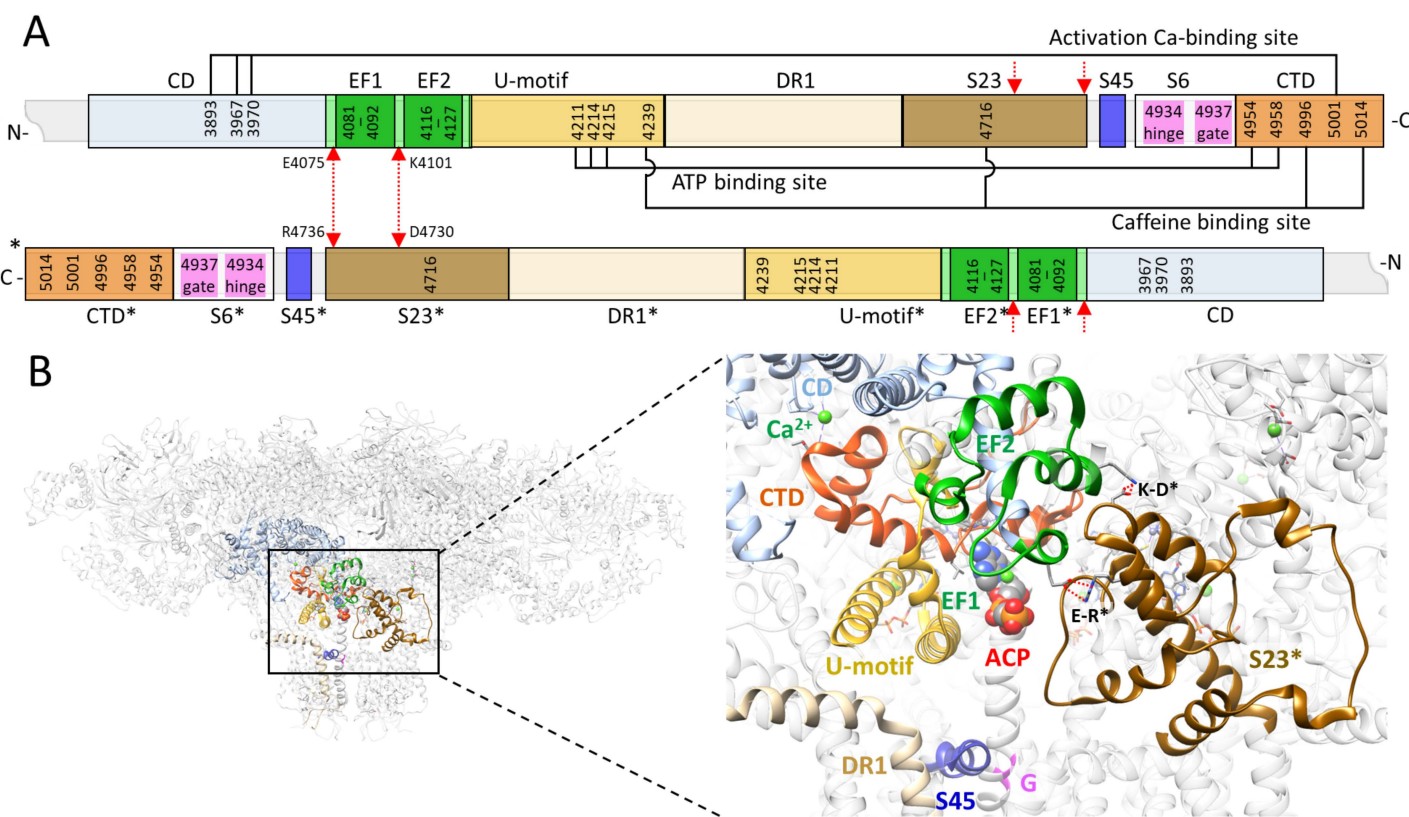

**Fig 1. The sequences and the molecular structures related to the RyR channel operation. (A)** The block scheme of the rRyR1 C-terminal quarter (3667-5037) for two neighbor monomers. Light blue - the Central domain (CD); light green – the EF-hand region; dark green – EF1 and EF2 motifs; tan – the Divergent region 1 (DR1); yellow – U-motif; light brown – S23 loop; blue – S45 linker (S45); white – the S6 segment with the hinge and gate residues in magenta; orange – the C-terminal domain (CTD). Red arrows – salt bridges between interacting residues of the EF-hand region and the S23* loop. The asterisks indicate a reference to the neighbor RyR monomer located counterclockwise in the top view of the channel. The connectors connect the canonical amino acid residues of the indicated binding sites. **(B)** Left – the see-through side view of a RyR1 tetramer (Ca-inactivated state, rabbit 7tdg [22]); top – the cytoplasmic side, bottom – the SR luminal side. Right – the detail of regions involved in the RyR operation. Colors correspond to A. Green spheres – Ca atoms at the activation binding sites, and at the ACP (adenosine 5′-[β,γ-methylene]triphosphate) molecule bound at the ATP site (center). The brown color indicates the S23* loop of the neighbor monomer. The short red dotted lines correspond to the salt bridges between residues K4101-D4730* and E4075-R4736* (red arrows in A). G - the hinge and gate residues D4934 and I4937 of the S6 segment.

low-magnesium conditions. The RyR1 structures 7tdg, 7tdi, 7tdj, and 7tdk determined at 3.7 mM $Ca^{2+}$ were proposed to represent the inactivated state since they showed the conduction pore closed (diameter of ≈11 Å at Cα atoms of I4937) [22]. Importantly, the 7tdh structure obtained in parallel under the same conditions showed the pore open (diameter of 14.3 Å) [22] indicating that RyR may attain two different states under the same conditions. This could explain in part why the RyR2 structure 6jiy obtained at 5 mM $Ca^{2+}$ had the pore open (diameter of 17 Å) [10], at odds with the expectation to find the channel in the inactivated state with a closed pore at so high $Ca^{2+}$ concentration. Confoundingly, even at 3.7 mM $Ca^{2+}$ or 5–10 mM $Mg^{2+}$ concentrations in the reconstitution media the Ca or Mg atoms were observed only at the Ca-activation site, near the bound ATP molecule, or in the ion flux pathway of RyR1 but not elsewhere [22,23]. This indicates that the potential inhibition site has an affinity for divalent ions smaller than ATP ($K_d$ ≈200 µM, [45]).

Another region that seems to play a role in divalent-ion-dependent inactivation is the S23 loop. Three amino acids (R4736, Y4733, G4732) of the S23 loop were proposed as important components of $Ca^{2+}$-dependent inhibition in rabbit RyR1 [37]. Their mutation induces malignant hyperthermia (MH) and increases the IC50 for calcium-dependent inhibition. Indicatively, the S23 loop was found nearer the EF-hand region of the adjacent monomer in the RyR1 structure obtained at 10 mM $Ca^{2+}$ than in the structure at <0.1 µM $Ca^{2+}$ [46] and they were connected by salt bridges in the inactivated RyR1 structure but not in the open RyR1 structure determined under the same conditions [22]. Taking together the findings mentioned above, it may be concluded that the EF-hand region is a serious candidate for the RyR inhibition site; however, a functional or structural explanation of the differences between the RyR1 and RyR2 inhibition was not revealed. Part of the problem is in the relatively low resolution of the EF-hand region in cryo-EM structural studies.

Another RyR region that was shown to participate in the regulation of RyR activity is the S45 linker (amino acid residues 4820–4833 in rRyR1). Alanine substitution of residues T4825 or S4829 in RyR1 decreased its sensitivity to caffeine, the maximum of the open probability, and the maximum of ryanodine binding, while the mutation L4827A and the MH-linked mutation T4825I increased both maxima [47]. Moreover, the MH mutations T4825I and H4832Y were reported to increase the IC50 of RyR1 to inactivation by $Ca^{2+}$ [37]. Thus, this segment plays a role in both the activation and the inactivation of the ryanodine receptor.

Most of the published RyR structures were collected under experimental conditions that vary between laboratories. As a result, the electron density maps vary and deductions inferred from individual maps and structural models may not be equally valid for particular aspects. Similar problems occurred with recordings and interpretation of functional experiments. The question arises as to whether these are even compatible. If yes, it should be possible to construct a functional model from structural data that would explain functional data. To this aim, we selected a large set of published RyR1 and RyR2 structures determined under conditions comparable with relevant single-channel experiments and estimated relationships between divalent ion binding and signaling pathways. We focused on the allosteric mechanism of RyR control by divalent ions, with the use of bioinformatic and structural analysis methods. We compare RyR1 and RyR2 structures and construct a fundamental allosteric mechanism common for both RyR isoforms. The proposed mechanism is verified by an operational model of the RyR channel open probability based on the MWC theory [48] and statistical mechanics [49].

## Results

To correlate the structure of RyR with its function we took advantage of published RyR structures and RyR open probabilities obtained under similar conditions. RyR structures selected for analysis were obtained under conditions stabilizing the closed, primed, open, or inactivated structural states. RyR2 structures of the inactivated state have not been published yet. Open probabilities of both RyR1 and RyR2 selected for analysis were determined at $Ca^{2+}$ and $Mg^{2+}$ concentrations causing activation and inhibition of RyRs. All structures and open probabilities analyzed here are listed in the S1 File and in the Fig 10 worksheet of the S4 File, respectively.

Below we first examine the molecular structure of the EF-hand pair EF1/EF2, the candidate inhibition site, for its propensity to bind divalent ions. Second, we analyze the spatial interaction between the EF-hand region and the S23 loop,

the proposed inactivation path. Third, we evaluate the relative positions of the RyR central core domains relevant to the RyR single-channel activity regulation, in all examined RyR states and isoforms. Fourth, we identify the allosteric network in both RyR isoforms from the regulatory allosteric sites to the channel gate for the representative RyR states. Fifth, we construct a comprehensive model of RyR operation and use it to calculate the dependences of RyR1 and RyR2 open probabilities on the divalent ion concentration. Finally, we compare the model predictions with the relevant single-channel open probability data.

**The propensity of EF1 and EF2 loops to bind divalent ions**

In all RyR structures, EF1 and EF2 have the form of a typical EF-hand pair, similar to Ca-binding sites of specialized Ca-binding proteins. To evaluate and compare their ability to bind divalent ions we used the identity score (IS) of EF-hand loops (see Methods), which has a maximum value of 12/6/3 (21) for calmodulin. The sequence of the EF1 loop in RyR1, VTDPRGLISKKD, the same in all examined mammals (rabbit, rat, mouse, human, dog, and pig), has IS of 10/4/2 (16). The corresponding EF1 loop sequence for RyR2 is DPDGKGIISKRD for the rabbit and DPDGKGVISKRD for the other examined mammals, both having IS of 11/5/3 (18), by 2 points higher than for the EF1 of RyR1. The sequence of the EF2 loop is EADENEMINCEE in human, mouse, and rat RyR1; EADENEMINFEE in rabbit RyR1, and EADENEMIDCEE in pig RyR1, while in RyR2 it is ETDENETLDYEE in all examined species. All EF2 sequences have an IS of 10/5/2 (17). The identity score of the scrambled RyR1 EF1 sequence YDGDKPSKLDIT [50] is 9/4/2 (15), that is, by 1 point lower than that of EF1 in WT RyR1. The scores of EF1 and EF2 are comparable with the calmodulin EF-hand sequence score of 12/6/3 (21) suggesting that both the EF1 and EF2 loops and even the scrambled EF1 loop can bind divalent ions but probably with a reduced affinity and selectivity.

We examined the theoretical ion binding ability of EF1 and EF2 hands in the region corresponding to rRyR1 residues 4063–4196, which starts ahead of the EF-hand region and ends behind the U-motif onset, to keep the secondary structure intact. This region was extracted from 10 recent RyR1 and RyR2 structures specified in the S1 File. The scrambled versions of EF1 were created using RyR structures determined at high $Ca^{2+}$ concentration [22], in which the wild-type EF1 sequence of amino acids 4079–4092 was replaced by the scrambled sequence YDGDKPSKLDITRV that showed RyR2-type inactivation in experiments [50]. Finally, all structures corresponding to rRyR1 residues 4063–4196 were subjected to energy minimization and submitted to the MIB2 server [51] for evaluation of the ion binding score (IBS) of individual amino acid residues and the number of ion binding poses (NIBP) for Ca and Mg ions. The server determines the IBS values of individual residues using sequence and structure conservation comparison with 409 and 209 pre-set templates from the PDB database for $Ca^{2+}$ and $Mg^{2+}$, respectively [52], and by assessing the similarity of the configuration of the residue to its configurations in known structures of its complexes with the given metal [53]. The server determines potential ion-binding sites by locally aligning the query protein with the metal ion-binding templates and calculating its score as the RMSD-weighted scoring function Z. The site is accepted if it has a scoring function Z>1, and based on the local 3D structure alignment between the query protein and the metal ion-binding template, the metal ion in the template is transformed into the query protein structure [52]. The larger the IBS value, the larger the tendency of the residue to bind the ion. The larger the NIBP value, the larger the number of such complexes with an acceptable structure. The IBSs were estimated for amino acid residues 1, 3, 5, 7, 9, and 12 of the EF1 and EF2 loops involved in ion binding. As a comparison baseline, we used the residues of 4063–4196 without the 12 metal-binding residues. The number of IBPs was determined as the number of all poses that involved at least 2 residues of the ion binding loops. Table 1 presents statistics on IBS and NIBP data.

The IBS values of the ion-binding residues of both EF1 and EF2 were significantly larger than the IBS values of the remaining residues (KWANOVA, p < 0.001). ANOVA analysis indicated that IBS of EF1 for $Ca^{2+}$ ions was significantly higher in RyR1 and RyR2 structures than in the scrambled RyR1 structure RyR1scr (p = 0.0128 and p < 0.001) but for $Mg^{2+}$ ions only the difference between RyR2 and RyR1scr was significant (p < 0.001). EF1 and EF2 bind $Ca^{2+}$ equally in both isoforms. In RyR2, EF2 binds $Mg^{2+}$ more strongly than EF1 (p = 0.0028) while in RyR1 both EF hands bind $Mg^{2+}$

**Table 1. The ion-binding ability of EF1 and EF2 in RyR isoforms.**

| Structure | N | IBS (mean ± SEM) | | | | | | NIBP (mean ± SEM) | | | |
|---|---|---|---|---|---|---|---|---|---|---|---|
| | | EF1 | | EF2 | | Baseline | | EF1 | | EF2 | |
| | | Ca | Mg | Ca | Mg | Ca | Mg | Ca | Mg | Ca | Mg |
| RyR1 | 10 | 0.7±0.1 | 0.4±0.1 | 0.8±0.2 | 0.8±0.1 | -0.07±0.03 | -0.06±0.03 | 1.9±0.4 | 0.7±0.3 | 4.3±1.4 | 2.4±0.8 |
| RyR2 | 10 | 1.2±0.1 | 1.1±0.1 | 1.5±0.2 | 1.7±0.1 | -0.14±0.03 | -0.13±0.03 | 6.6±1.8 | 2.1±0.4 | 9.3±2.3 | 5.1±1.4 |
| RyR1scr | 7 | 0.2±0.1 | 0.2±0.1 | 0.7±0.1 | 0.7±0.1 | -0.04±0.03 | -0.04±0.03 | 0.7±0.4 | 0.9±0.3 | 2.1±0.7 | 1.7±0.6 |
| CaM-N | 1 | 2.3 | 2.2 | 2.3 | 2.2 | -0.42 | -0.41 | 56 | 8 | 42 | 5 |
| CaM-C | 1 | 1.6 | 2.0 | 1.8 | 1.8 | -0.37 | -0.39 | 8 | 4 | 37 | 7 |

IBS – ion binding score. NIBP – ion binding poses. N – the number of evaluated RyR structures. RyR1scr stays for the RyR1 with scrambled EF1-hand sequence. CaM-N and CaM-C stay for the N-terminal and C-terminal EF-hand pairs of calmodulin, respectively. The values for calmodulin were estimated using the structure 1exr [54].

equally. Importantly, however, the EF1 had an IBS for $Ca^{2+}$ and $Mg^{2+}$ significantly larger in RyR2 than in RyR1 (p = 0.0011 and 0.0014, respectively). The extent of these differences can be assessed by comparison with calmodulin (Table 1), which also possesses two pairs of EF-hands that bind $M^{2+}$, albeit with micromolar affinity. The lower IBS values of EF-hands in RyRs are consistent with the observed sub-millimolar concentration range of inhibitory potency of $Ca^{2+}$ and $Mg^{2+}$ in RyR isoforms.

The estimates of NIBPs (Table 1) emphasize findings on IBS of the RyR1 and RyR2 EF-hands, regarding their sensitivity to the type of divalent ion. As demonstrated, the NIBP of CaM-N and CaM-C for $Ca^{2+}$ is markedly higher than that for $Mg^{2+}$, in agreement with the known selectivity of calmodulin for these ions. In both EF1 and EF2 of RyRs, the difference in NIBP of $Ca^{2+}$ and $Mg^{2+}$ is small, indicating low selectivity for divalent ions. Comparison of NIBP in RyR isoforms indicates a better binding of $M^{2+}$ in RyR2 than in RyR1. The structures with scrambled EF1 loop returned lower values of IBS and NIBP.

The lower IS, IBS, and NIBP values of the scrambled RyR1 structures are in agreement with the decreased $Ca^{2+}$ inhibitory potency of the scrambled EF1-hand RyR1 construct [50]. The higher IBS and NIBP for $Ca^{2+}$ and $Mg^{2+}$ of the EF1-hand in RyR2 than in RyR1 is at odds with the weaker inhibition of ³H-ryanodine binding by $Ca^{2+}$ ions in RyR1 channels with their EF1 motif exchanged for the RyR2 counterpart [55], suggesting that the effect of this exchange cannot be caused by the higher $Ca^{2+}$ affinity of the EF1 loop in RyR1 than in RyR2.

The EF2 of RyR2 has the same IS but somewhat higher IBS and NIBP values than the EF2 of RyR1. This theoretical result is in line with the lack of the effect of the exchange of the RyR1 region around the EF2 for the corresponding RyR2 region on the $Ca^{2+}$-dependent inhibition of ryanodine binding [55], meaning that the difference between Ca-dependent inactivation of RyR1 and RyR2 is not caused by the properties of the EF2 hand.

To summarize, the theoretical ion binding results provided no support for the hypothesis that the difference between RyR isoforms comes from the different affinity of their putative inhibitory sites to divalent ions. On the contrary, they indicate a substantial ability of EF1 and EF2 to bind divalent ions, a prerequisite to constitute the inhibitory Ca-binding site.

## Spatial interactions between the EF-hand and S23* regions

As we reasoned above, the difference in the propensity of the EF-hand region to bind divalent ions cannot explain the difference in the Ca-dependent inactivation between RyR isoforms. The question is then whether the difference can reside in the transmission of the ion-binding signal from the EF-hands to the channel gates. A clue was provided by Gomez et al. [37] who found that malignant hyperthermia mutations of three amino acids of the S23 loop shifted the inhibition of ryanodine binding in RyR1 to a higher $Ca^{2+}$ concentration, close to that of the RyR2, and proposed that these three amino acids are involved in the inter-monomeric interaction of the S23* loop with the EF-hand region [37]. Nayak and Samsó [22]

measured the distance between the EF-hand region and the S23* loop as the distance between the Cα atoms of pairs of residues allowing direct interaction at sufficient proximity. These included glutamate E4075 to arginine R4736* (distance D1) and lysine K4101 to aspartate D4730* (distance D2) in rabbit RyR1 structures. Both distances showed a tendency to decrease among RyR1 structures in the order closed (7k0t)> open (7tdh)> primed (5taq) ≈ Ca-inactivated (7tdg), indicating that these two pairs of residues may play a role in RyR1 inhibition [22]. We measured both distances in the whole set of RyR structures to examine the difference between the RyR isoforms.

The D1 distances measured between Cα atoms of the E4075-R4736* pair are summarized in Fig 2, and compared with the theoretical estimate of the maximal distance allowing their interaction. The median of D1 values measured in the RyR1-C structures is too large for the residues to interact, while those of the open, primed, or inactivated RyR1 structures are in the range allowing for proper interaction between residues in a suitable rotamer configuration. Of interest is the RyR1-Mg structure 7umz, defined as "ACP/10mM free Mg²⁺-closed state" [23], obtained at a very high concentration of Mg²⁺ ions that are supposed to stabilize the inactivated state [35]. This structure provided D1 similar to those found in primed, open, and inactivated states. The D1 measurements in RyR2 structures provided completely different results. In all RyR2 states the median D1 is substantially larger than that in RyR1 and, importantly, also larger than the maximum interaction distance estimated for hRyR2 D4030-R4665* (the equivalent of rRyR1 E4075-R4736*).

Measurements of D2 distances yielded only subtle distinctions between RyR structures (Fig 2, right). In all RyR1 structures, the median D2 values were neither substantially nor statistically different. In all RyR1 states, the K4101-D4730* separation allowed for proper residue interaction by H-bonds. In RyR2 structures, the D2 medians of individual states were only slightly larger than those in the respective RyR1 structures, much less so than in the case of D1 distances. The D2 medians of closed, open, and primed RyR2 states were similar and in the theoretical H-bond interaction range.

These results indicate that the glutamate-arginine* (E4075 and R4736*) distance D1 between the EF-hand and S23* regions is too large in all RyR2 structures, in contrast to the RyR1 primed, open, and inactivated states, in which the residues may potentially interact and transmit the binding signal. On the other hand, the D2 distances in all RyR1 and RyR2 structures assume values that may allow interaction of the lysine with the aspartate* (K4101-D4730*) in RyR1 or with the glutamate* in RyR2, and thus potentially participate in signal transmission.

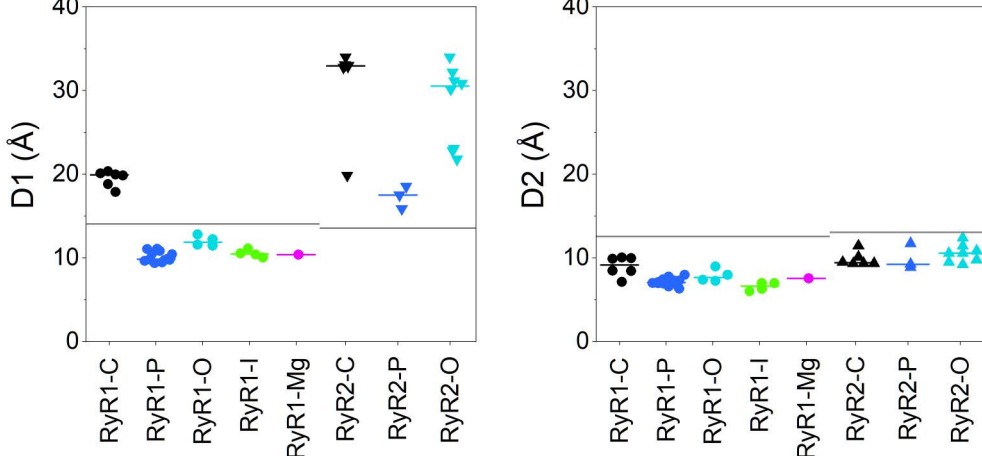

**Fig 2. Distances between key amino acid pairs of the EF-hand and S23* in RyR structures.** Left – Distances D1 between the Cα atoms of E4075 and R4736* or equivalent. Right – Distances D2 between the Cα atoms of K4101 and D4730* or equivalent. Circles – RyR1 states. Triangles - RyR2 states. C – closed state, black symbols. P – primed state, blue symbols. O – open state, cyan symbols. I – inactivated state, green symbols. Mg – high-magnesium state, magenta symbols. Short lines – the median values of groups. Long lines – the theoretical maximum distances allowing H-bond formation.

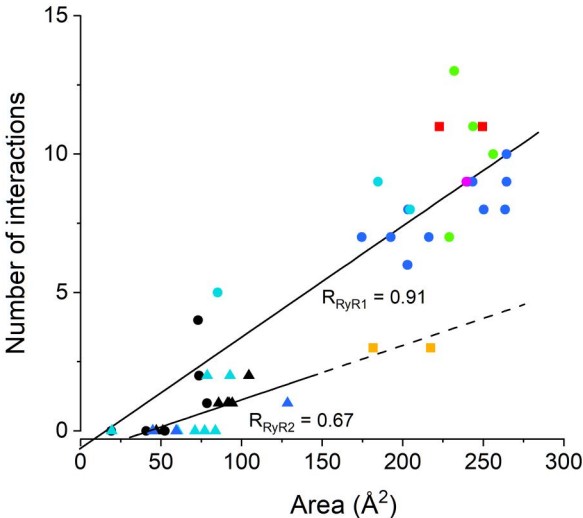

The resolution of published cryo-EM RyR structures is typically about 3–4 Å, which limits conclusions based only on estimates of specific interaction distances. Moreover, we cannot exclude the presence of interactions among other residues. A more robust measure of domain interactions could be the contact area between the juxtaposed surfaces of the EF-hand region and the S23* loop, and the number of possible electrostatic interactions (H-bonds and salt bridges) between their residues on the same set of RyR1 and RyR2 structures. For this purpose, the motifs corresponding to rabbit RyR1 residues E4032 – E4182 of one monomer, and to rabbit RyR1 residues D4695 - L4745 of the neighboring counterclockwise monomer were cut out from the corresponding RyR structures, optimized, and submitted to the PDBePISA server (see Methods for details). The results, summarized for all analyzed structures in Fig 3, point to a substantially larger contact area and number of interactions in primed, open, and inactivated RyR1 structures than in closed RyR1 and closed, primed, and open RyR2 structures, in line with the distance measurements presented above. The RyR1-Mg structure 7umz [23] provided both of the values similar to those found in primed, open, and inactivated states. The contact area and the number of electrostatic interactions correlate more steeply in RyR1 than in RyR2 structures. This indicates that the spatial arrangement of the two domains in RyR2 structures does not support the formation of electrostatic interactions as efficiently as in RyR1 structures, even if the two domains are near each other.

After optimization, 24 out of 27 RyR1 structures formed H-bonds and/or salt bridges between at least one of the residue pairs Q4100-I4731*, (E4075/D4079)-R4736*, (Q4100/K4101)-D4730* in all RyR1 states. The maximum number of simultaneous hydrogen bonds detected by the PDBePISA server was 6. On the other hand, in the corresponding RyR2 structures only 8 out of the 17 analyzed structures formed interactions between EF-hand – S23* residue pairs, with a maximum of two hydrogen bonds in a single structure. Moreover, the EF-hand aspartate and glutamate (corresponding to E4075 and D4079 in RyR1) could not form contact with the arginine corresponding to rRyR1 R4736*, the key S23* residue, in any RyR2 structure. These data suggest that interactions between the basic arginine residue R4736* and the acidic residues E4075 and D4079 are specific for Ca²⁺-dependent inactivation in RyR1, whereas the interactions between the lysine K4101 and the residues D4730* and I4731* (rRyR1 notation) may play a part in the inactivation of both RyR1 and RyR2 isoforms.

**Fig 3. Number of interactions and area between the EF-hand region and S23* loop in RyR structures.** Circles – RyR1 structures; Triangles – RyR2 structures; Black – closed state; Blue – primed state; Cyan – open state; Green – inactivated state; Magenta – high-Mg state. Orange squares – inactivated RyR1 with MH mutations R4736W, R4736Q; Red squares – inactivated RyR1 with MH mutations G4733E, F4732D. $R_{RyR1}$, $R_{RyR2}$ – Pearson's correlation coefficients for RyR1 and RyR2 data groups (solid lines); $n_{RyR1}$ = 27; $n_{RyR2}$ = 17; $p < 0.005$ in both cases). Note: the correlations were calculated without the data for mutated structures (squares). The dashed line extrapolates the correlation line.

Interestingly, the replacement of arginine R4736* for tryptophan or glutamine in RyR1 that alleviates RyR1 inhibition by $Ca^{2+}$ [37] provided a reduced number of interactions per interaction area (Fig 3, orange squares), similar to that predicted for RyR2 type interaction. However, replacements of glycine G4733* for glutamate and phenylalanine F4732* for aspartate, also known to alleviate RyR1 inhibition by $Ca^{2+}$ [37] did not affect the number of interactions (Fig 3, red squares). This means that the electron donor residue arginine R4736* is probably important for the transmission of the divalent ion binding signal from the EF-hands to the S23* loop in RyR1. On the other hand, the mutations G4733E and F4732D, both introducing a negative charge on S23, may act by neutralizing the positive charge of R4736 thus making the transmission less effective without changing the number of possible interactions.

## Relative positions of the RyR core domains

The above distance analyses suggested clearly that the relative positions of the EF-hand and S23* loop are state- and isoform-specific, and thus may affect the propensity of RyR1 and RyR2 for $M^{2+}$-dependent inactivation. Additional contributions to the inactivation could arise from other domains involved in the signal transmission that may assume different relative positions and shapes in specific RyR states and isoforms. Previous RMSD (root mean square displacement) analysis of the backbone Cα atoms revealed that the position of the C-terminal domain relative to the Central domain was shifted in primed and open RyR1/RyR2 structures from the position in closed structures [8,22]. An even higher degree of the C-terminal domain shift was present in the inactivated RyR1 structure 7tdg [22].

We inspected differences in conformational states of core domains located along the C-terminal quarter of the RyR monomer by means of the RMSD analysis using the same set of RyR structures as analyzed above. To begin with, the analyzed RyR structure and the reference RyR structure were aligned by their Central domains, selected as the aligning reference for sufficient size and well-defined state-independent structure. The distances between the corresponding Cα atoms of domains of interest in the tested and the reference structures were then used to calculate their $RMSD_{100}$ values. The graphs of Fig 4 compare $RMSD_{100}$ estimated relative to the closed RyR1 structure 7k0t (horizontal axis) and the inactivated RyR1 structure 7tdg (vertical axis). The S45 linker, known to participate in RyR gating, could not be analyzed in this way since its length of 13 residues is insufficient for correct $RMSD_{100}$ determination [56]. The $RMSD_{100}$ of domains in the same state and isoform were nearly always spread within a circle with a radius of 1 Å for all RyR structures. Thus, we considered the structure of a RyR region to differ between channel states if their mean $RMSD_{100}$ values differed by more than 2 Å [56].

**Comparison of domain positions in RyR1 and RyR2 structures.** The $RMSD_{100}$ values of the Central domain in all RyR structures were similar and independent of the RyR state or isoform (Fig 4A), as required for the self-aligned reference domain.

The C-terminal domain and U-motif (Fig 4B and 4C) of all RyR1-C and RyR2-C structures (black symbols) showed similar $RMSD_{100}$ values close to those of the reference closed structure (7k0t). The RyR1-I structure 7umz, obtained in the presence of high $Mg^{2+}$ concentration, showed $RMSD_{100}$ values halfway between those of closed and inactivated RyR1 structures.

In almost all other RyR1 and RyR2 structures, RMSDs of these domains were similar and close to that of the reference inactivated structure (7tdg). This confirms the displacement of the CTD and U-motif relative to the Central domain in primed, open, and inactivated RyR1 structures, observed previously by Nayak and Samso [22] for the open 7tdh, primed 7tdj, and inactivated 5taq RyR1 structures. Moreover, it reveals qualitatively and quantitatively similar rearrangements from the closed to the primed or open states in both the RyR1 and RyR2 isoforms, indicating their similar responsiveness to the activation and inactivation conditions in both RyR isoforms.

The EF-hand domains of RyR1 and RyR2 structures displayed markedly dissimilar RMSDs. In RyR1, the open, primed, and inactivated structures (colored circles) were notably different from the closed but very close to the inactivated reference (RyR1) structure. The structure 7umz, obtained at a high $Mg^{2+}$ concentration, showed $RMSD_{100}$ values significantly

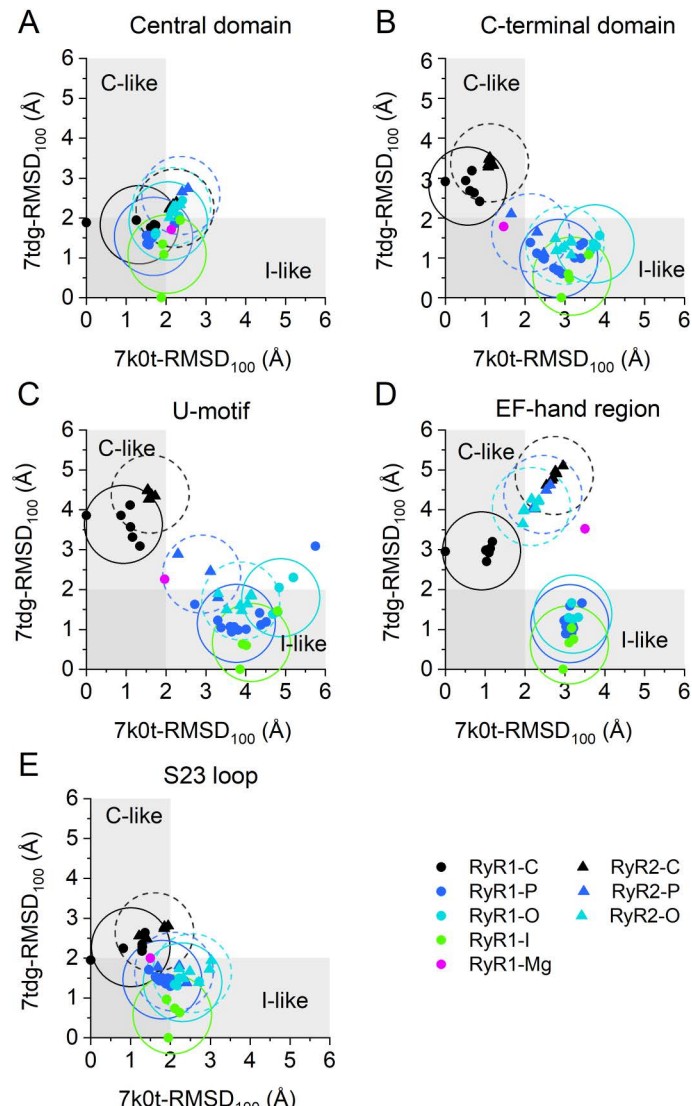

**Fig 4. The RMSD analysis of RyR structures.** Horizontal axis – RMSD$_{100}$ of the RyR structures relative to the closed RyR1 structure 7k0t. Vertical axis – RMSD$_{100}$ of the RyR structures relative to the inactivated RyR1 structure 7tdg. The circles of 1 Å radius are centered on the mean RMSDs of the corresponding group of RyR isoforms and states and appraise the significance of differences between groups (when the corresponding circles do not intersect the difference is significant). The solid circle data points and the solid circle lines indicate RyR1 data; the solid triangle data points and the dashed circle lines indicate RyR2 data. Colors indicate RyR states (see inset). C-like and I-like grey bands indicate the similarity of RMSD$_{100}$ values to either the closed or inactivated reference states.

different from both the closed and inactivated reference RyR1 structures. In RyR2, the EF-hand region of all structures (triangles) also displayed notably different RMSD$_{100}$ values relative to either of the reference (RyR1) structures (Fig 4D). These data reinforce the evidence of the structural differences between the EF-hands of RyR1 and RyR2 isoforms indicated by the distance analysis above.

The RMSD values of the S23 loop in all RyR structures were low, almost the same, and similar to those of both the closed and inactivated reference structures. This points to the very high structural stability of the S23 loop within the RyR monomer relative to the Central domain.

**Comparison of domain positions in different RyR states.** In all domains except the EF-hand, the RMSDs of the closed RyR1 and RyR2 structures were very similar irrespective of the isoform. The RMSDs of the primed, open, and inactivated RyR1 structures and the open and primed RyR2 structures were also very similar but markedly different from those of the closed RyR1 and RyR2 structures. However, while there is a pronounced difference between the mean $RMSD_{100}$ of the EF-hand in closed RyR1 structures on the one hand and the open, primed, and inactivated RyR1 structures, the differences between the closed, open, and primed RyR2 structures are not significant. This pronounced distinction in the EF-hand position between RyR1 and RyR2 indicates that the inactivation signal in RyR2 may have a lower chance of passing to the neighbor monomer. These findings also indicate that in RyR1 channels, the structural prerequisites for the inactivation signaling are present also in primed and open channels and that the binding of divalent ions to the inhibition site at the EF-hand region may constitute the inactivation signal.

Whereas the relative positions of all five examined domains were similar in the primed, open, and inactivated states (Fig 4), these states differed by their flexion angle (see Methods). Despite the large dispersion, the medians of flexion angles showed a tendency to decrease in both RyR1 or RyR2 structures in the order: closed state ≥ primed state ≥ high $Mg^{2+}$ state ≥ open state ≥ inactivated state, in RyR1; closed state ≥ primed state ≥ open state, in RyR2 (see S1 File for details).

**Interaction of EF-hand region with S23 loop.** To better understand the above-disclosed difference in the displacement of the EF-hand region and S23 loop between RyR1 and RyR2 structures (Fig 4D and 4E), we aligned the open RyR1 and RyR2 model structures on their more static S23* loop (Fig 5). We compared here only the open states, since according to the distance analysis (Fig 2) the distances in the closed states are too large, the inactivated RyR2 structure is not available, and the distances D1 and D2 are similar in the open and inactivated RyR1. A close look at the structure of the EF-hand region (Fig 5A) reveals substantially different conformations of the two isoforms in the segment starting just before EF1 and ending just after EF2 (residues 4052–4140 in rRyR1). In RyR2, the backbones of both EF1 and EF2 are diverted downward relative to their counterparts in RyR1. Consequently, the D4030 of RyR2 (the equivalent of E4075 in RyR1) withdraws from interacting with S23* (Fig 5B and 5C). A similar change in orientation of the EF-hand relative to the S23 loop was observed in all primed and open RyR2 structures examined (see S1 Fig for the visual comparison of all evaluated primed, open, and inactivated RyR1 and RyR2 structures). This confirms the results of the distance analysis, namely that the formation of hydrogen bonds between K4101 and D4730* and between E4075 and R4736* is possible in primed, open, and inactivated RyR1 structures, while the formation of a hydrogen bond is possible only between K4056 and E4659* (hRyR2 equivalents of K4101 and D4730*) but not between D4030 and R4665* (hRyR2 equivalents of E4075 and R4736*) in the corresponding primed and open RyR2 structures. The difference stems most probably from the low sequence identity (52%) for the EF-hand region (rRyR1 4071–4137 and hRyR2 4026–4092). This is much less than the sequence identity of the adjacent U-motif (81%) and the Central domain (82%), at which RyR1 and RyR2 are aligned much better (Fig 5A).

## Allosteric pathways from the ion-binding sites to the gate

Transmission of a signal from the regulatory site to the active site should proceed by propagation of a perturbation through a sequence of contacts between nearby residues. This principle is used in the OHM server for the detection of allosteric pathways [57]. Previously, analysis of such pathways in RyR1 structures revealed how the cytosolic calcium, ATP, and caffeine activate RyR1 [58]. Here we have used the OHM server to explore the regulatory network of both ion-dependent allosteric pathways. We will use the term "activation pathway" for the pathway commencing at the high-affinity $Ca^{2+}$-binding site, and "inhibition pathway" for the pathway commencing at the low-affinity, low-selectivity $M^{2+}$-binding site associated with the inhibition site. Both will be considered in the model of RyR operation. The term "residue importance" defines the extent to which the given residue is involved in the propagation of a perturbation from the allosteric site to the active site, i.e., the fraction of simulated perturbations transmitted through this particular residue. The more contacts are present between two residues, the larger is the probability that a perturbation will be propagated from one to the other residue.

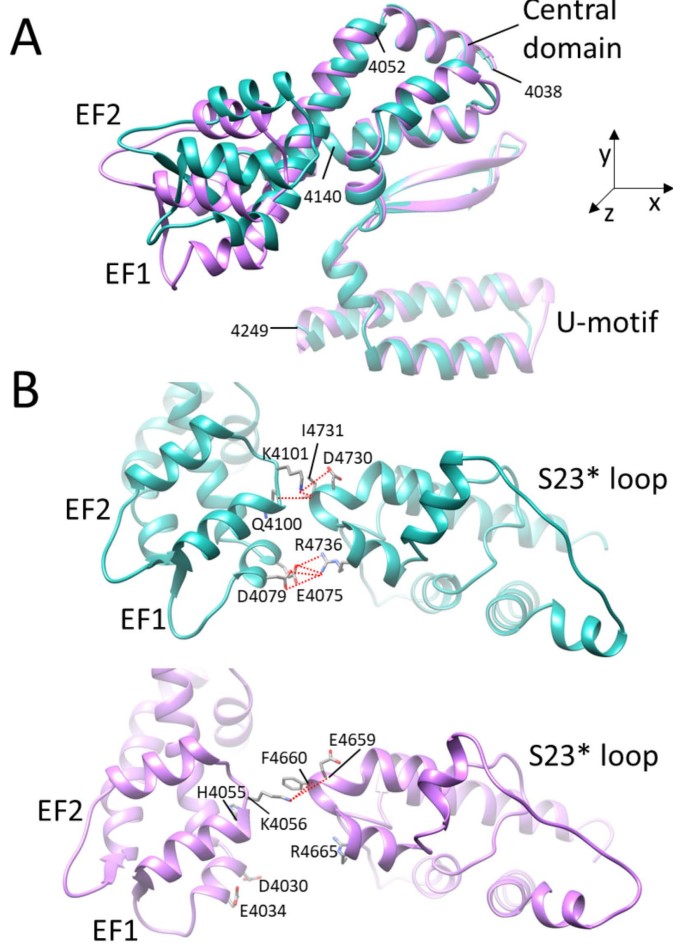

**Fig 5. Possible interactions between the EF-hand and S23 in the open RyR1 and RyR2 structures. A** – Overlay of the EF-hand region in the open RyR1 (7m6l, cyan) and RyR2 (7ua9, magenta) structures aligned at their S23 loops (out of the image plane). Note the differences in the EF1 and EF2 positions and almost the same positions of the Central domain and U-motif segments. The RyR1 residue sequence numbers are given for orientation. **B** – Possible residue interactions between the EF-hand region (left) and S23* loop (right) in the open RyR1 (cyan) and RyR2 (magenta) structures. The structures in B are rotated by 30° counterclockwise about their x, y, and z axes relative to their orientation in A. Dotted red lines -– structurally possible H-bonds between the interacting residues. The spatial arrangement of all primed, open, and inactivated structures is compared for RyR1 and RyR2 in S1 Fig.

Since the allosteric analysis requires a complete and correct structural model, only RyR structures with the best resolution and the fewest missing residues were processed (see Methods). According to these criteria we selected five RyR1 model structures (Table 2): one closed (7k0t), one primed (7tzc), two open (7m6l and 7tdh), and one inactivated (7tdg); and four RyR2 model structures: two closed (7vmm and 7ua5) and two open (7vmp and 7ua9). The structures 7tdg and 7tdh represent two RyR1 state configurations in the same sample; similarly, the structures 7ua5 and 7ua9 represent two RyR2 state configurations in the same sample (see S2 File for the sequences and structures of the models).

In the selected structures, we identified residues with high residue importance (RI) for both the inhibition and activation network and compared them with residues important for ligand binding and gating of RyR (Table 3). These residues were then identified in individual paths and the paths containing identical high-RI residues were grouped. To increase the visibility of the main pathways, residues with RI less than 5% of the maximum RI for the given structure were not considered.

**Table 2. Model structures used for identifying the allosteric pathways.**

| Structure | 7k0t | 7tzc | 7m6l | 7tdh | 7tdg | 7vmm | 7ua5 | 7vmp | 7ua9 |
|---|---|---|---|---|---|---|---|---|---|
| **Species** | rabbit | rabbit | rabbit | rabbit | rabbit | mouse | human | mouse | human |
| **RyR type** | RyR1 | RyR1 | RyR1 | RyR1 | RyR1 | RyR2 | RyR2 | RyR2 | RyR2 |
| **State** | closed | primed | open | open | inactivated | closed | closed | open | open |
| **Ligands** | ACP | ATP CFF Ca | ATP CFF Ca | CaACP Ca | CaACP Ca | none | ATP | Ca | ATP XAN Ca |

ACP – adenosine 5′-[β,γ-methylene]triphosphate; ATP – adenosine triphosphate; CFF – caffeine; XAN – xanthine.

**Table 3. The list of important residues examined for contribution to allosteric networks.**

| Ligand binding sites | Residue numbers (rabbit RyR1 numbering)# |
|---|---|
| ATP binding site | **4211, 4214, 4215,** 4218, **4954, 4955,** 4957, **4958, 4959,** 4960, 4975, **4979,** 4983, **4984, 4985** |
| Caffeine binding site | **3753**, **4239**, 4242, 4671, **4715**, **4996**, 5011, **5014**, 5015 |
| **Interaction sites** | |
| EF-hand region | 4075, 4079, 4100, 4101 |
| S23* loop | 4730*, 4731*, 4736* |
| **Regulatory site** | |
| S45 linker | 4821-4834 |

The ATP and CFF binding sites were defined as a group of residues that formed contact with a ligand in at least one of the RyR structures listed in the S1 File, given that the position of a ligand in the structure is not well-defined [5] and that not all examined model structures contained a ligand at the binding site. The interacting residues of the EF-hand– S23* regions were those identified by the PDBePISA to form a salt bridge or an H-bond between neighboring monomers in at least one RyR structure (S1 File).

#Residue numbers in bold were originally defined as the ATP and CFF binding site-forming residues by [5].

For each RyR structure, this process precipitated in major pathways that shared residues of the regulatory ligand-binding sites, inter-monomeric interaction sites, and the regulatory site of the S45 linker (Table 3).

**The inhibition network.** In the inhibition network analysis, we defined the ion-binding loops of the EF1 and EF2 as the allosteric inhibition site (INH), and the hinge and gate of S6 and S6* as the active site (GATE). In the network output list of the OHM server (see S2 File), large RI values were observed at the ATP-binding residues of the U-motif and the C-terminal domain, in addition to residues next to the EF-hand loops and the GATE (Fig 6). In several model structures, RIs larger than the threshold value also occurred at several residues of the S45, S45*, and S23*segments.

Two major pathways denoted as the intra-monomeric and the inter-monomeric pathway were identified in the structural models of both RyR isoforms. Both pathways displayed several branches that partially fluctuated among model structures. Notably, most branches comprised residues either of the ATP binding site, known to support the calcium-dependent activation [5,58], or of the S45 linker, known to participate in the regulation of channel gating [37,47].

The intra-monomeric inhibition pathway was identified in all 9 investigated model structures and represented 48 – 100% of the first 100 inactivation paths generated by OHM for individual model structures (see S2 File). In both RyR isoforms, this pathway was formed by six major branches (I1 – I6, Tables 5 and S1) all passing through the U-motif and the ATP-binding site residues. Each RyR model structure contained at least one but up to three intra-monomeric branches. Individual branches, which occurred in various proportions in different model structures, differed in specific residues of the ATP-binding site. Branches I4 – I6 also contained residues of the S45 linker, more often in RyR1 than RyR2 isoforms, and more often in the closed pore structures (C, P, I) than in the open structures. Individual structures contained 1 – 3 intra-monomeric branches. Three RyR1 and two RyR2 structures contained only 1 intra-monomeric branch.

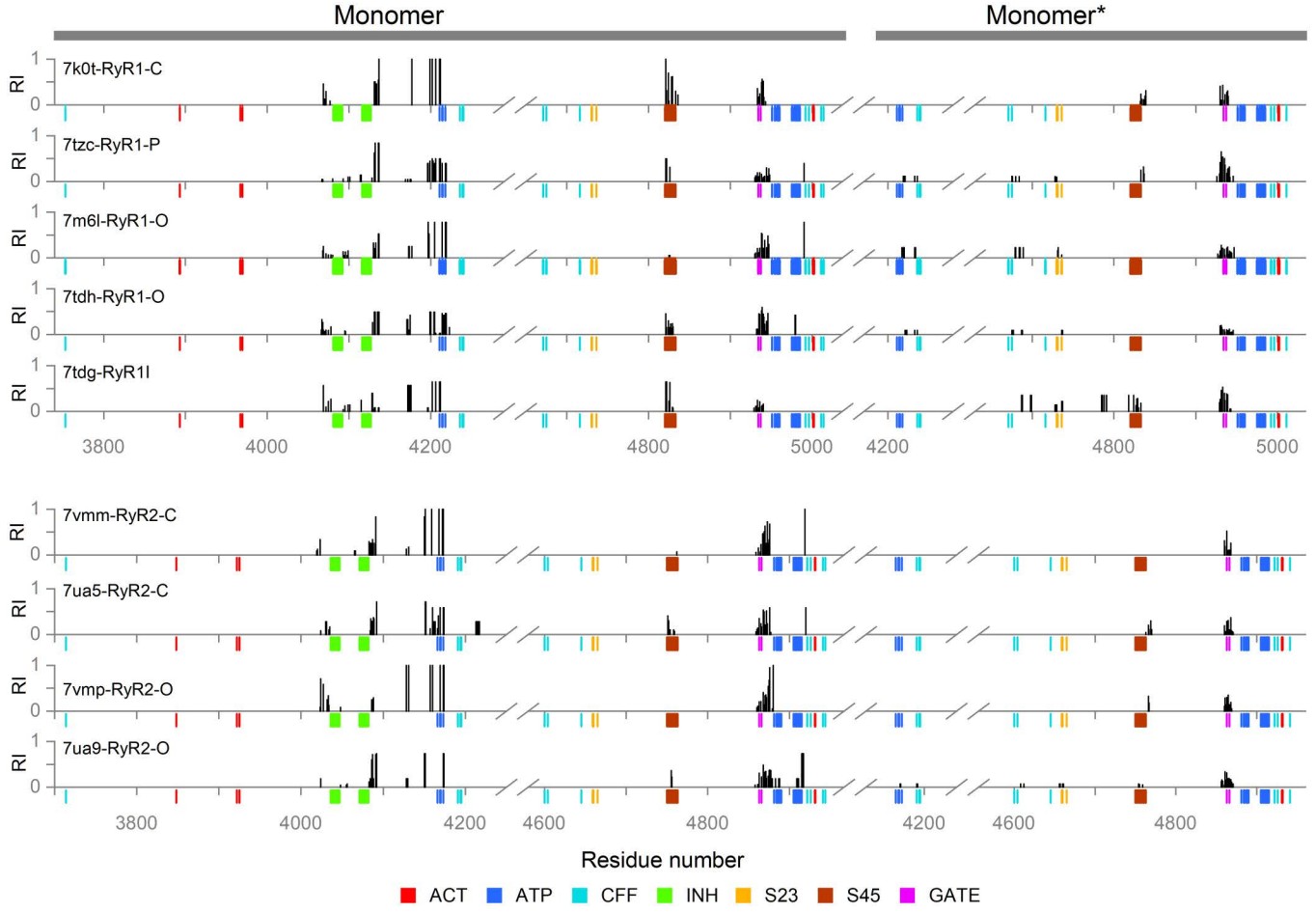

**Fig 6. The residue importance in the inhibition pathways.** The RI was calculated for all detected inactivation paths and mapped on the RyR sequences. The graph displays only RI > 0.05. The color-coded marks under the sequence axis indicate the positions of important residues from Tables 3 and 4, see the legend at the bottom, where: ACT – Ca²⁺-binding activation site, INH – M²⁺-binding inhibition site, ATP – ATP-binding site, CFF – caffeine/xanthine binding site, S23 – S23 loop, S45 – S45 linker, and GATE – channel gating residues.

The inter-monomeric pathway connecting the EF1/2 loops with the S23* loop was detected by OHM in 5 – 52% of the first 100 inhibition paths for individual model structures (Tables 5 and S1). This pathway diverged to 6 branches (I7 – I12), which all involved one of the S23* residues R4736*, D4730*, and I4731* or its hRyR2 equivalent F4660*, identical to those forming a salt bridge or H-bond with the EF-hand region as identified by the PDBePISA data (see above). Notably, this pathway was present in all RyR1 structures except for the RyR1-C but only in one RyR2 structure (Table 5). Individual RyR structures contained 1 – 2 inter-monomeric branches. In the primed and open RyR1 structures, this pathway participated by only 5 – 28%, but it was dominant by 52% in the inactivated RyR1 structure 7tdg. In the RyR2 open structure 7ua9 it participated by 8% (branch I10). Notably, four out of 6 inter-monomeric branches used the U*-motif: three in RyR1 and one in RyR2 structures. Two branches of the inactivated RyR1 structure 7tdg connected the S23* with the S45* linker directly and not through the U*-motif. The inter-monomeric pathway reached the gate *via* the S6* segment.

**The activation network.** In the activation network, we defined the Ca²⁺ binding residues of the CD and CTD as the allosteric activation site (ACT), and the hinge and gate residues of the S6 and S6* as the active site (GATE). The active

**Table 4. The residues of the allosteric and active sites (rRyR1 numbering).**

| Allosteric sites | Residues |
|---|---|
| INH: EF-hand loops | 4081-4092, 4116-4127 |
| ACT: Ca²⁺ activation site | **3893, 3967**, 3970, **5001**, 5002 |
| **Active site** | |
| GATE: S6 Hinge and Gate | G4934, G4934*, I4937, I4937* |

Residues with * belong to the monomer neighboring the monomer bearing the analyzed allosteric site.

Residue numbers in bold were originally defined as the Ca²⁺ binding site-forming residues by [5].

**Table 5. Fractional occurrence of branches in the inhibition network pathways.**

| Main branches connecting the INH and GATE sites | | Fraction of paths (%) | | | | | | | | |
|---|---|---|---|---|---|---|---|---|---|---|
| | | RyR1 | | | | | RyR2 | | | |
| | | C | P | O | O† | I† | C | C† | O† | O |
| | | 7k0t | 7tzc | 7m6l | 7tdh | 7tdg | 7vmm | 7ua5 | 7ua9 | 7vmp |
| Name | Intra-monomeric pathways | | | | | | | | | |
| I1 | -EF - U - ATP - S6- | | 42 | 72 | 60 | | 100 | 63 | | 100 |
| I2 | -EF - U - ATP - S6- | | | | | | | | 18 | |
| I3 | -EF - U - ATP - S6- | | | | | | | | 74 | |
| I4 | -EF - U - ATP - S45 - S6- | | 53 | | | 48 | | 24 | | |
| I5 | -EF - U - ATP - S45 - S6- | | | | 27 | | | | | |
| I6 | -EF - U - ATP - S45 - S6- | 100 | | | 4 | | | | | |
| | Subtotal | 100 | 95 | 72 | 91 | 48 | 100 | 88# | 92 | 100 |
| | Inter-monomeric pathways | | | | | | | | | |
| I7 | -EF - S23* - U* - S6*- | | | | 14 | 9 | | | | |
| I8 | -EF - S23* - U* - S6*- | | | 14 | | | | | | |
| I9 | -EF - S23* - U* - S6*- | | 5 | | | | | | | |
| I10 | -EF - S23* - U* - S45* - S6*- | | | | | | | | 8 | |
| I11 | -EF- S23*- S45*-S6*- | | | | | 40 | | | | |
| I12 | -EF- S23*- S45*-S6*- | | | | | 12 | | | | |
| | Subtotal | 0 | 5 | 28 | 9 | 52 | 0 | 0 | 8 | 0 |
| | TOTAL | 100 | 100 | 100 | 100 | 100 | 100 | 88# | 100 | 100 |

INH and GATE are defined in Table 4. The abbreviations EF, U, ATP, S23, S45, and S6 refer to the EF-hand region, U-motif, ATP-binding site, S23 loop, S45 linker, and S6 helix, respectively. These domains contribute by one or more residues to the path (see S1 Table for the list of critical residues of individual branches).

†Marks a pair of RyR1 and RyR2 structures obtained in the same experiment.

#The remaining 12% of the allosteric signal passed through the poorly resolved DR1 region.

site was the same as defined for the inhibition pathway analysis since no other gates are present in the RyR pore. The large RI values were observed at the ATP-binding site and the C-terminal domain residues near the CFF binding site, in addition to residues contacting ACT and the GATE (Fig 7). In several model structures, RIs larger than the threshold value also occurred at several residues of the S45 linker.

The activation network consisted of a single intra-monomeric pathway with nine branches running close to each other (Tables 6 and S2). All branches comprised the CTD, the U-motif, and the S6 helix. Six branches involved one or more residues of the ATP-binding site, two branches involved residues of the caffeine/xanthine binding site known to support the calcium-dependent activation [5,58], and three branches involved residues of the S45 linker known to participate in

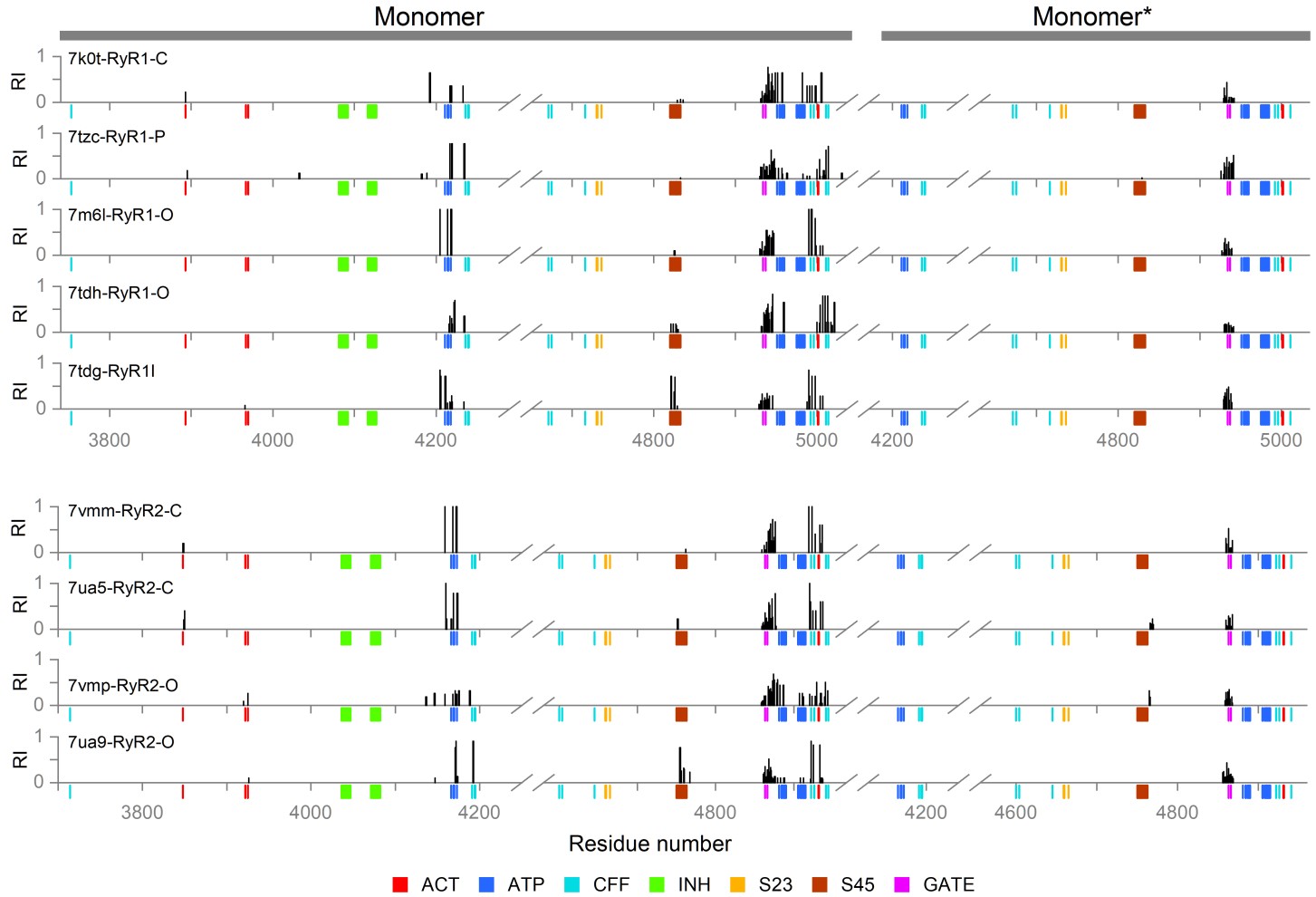

**Fig 7. The residue importance in the activation pathways.** The RI was calculated for all detected activation paths and mapped on the RyR sequences. The graph displays only RI > 0.05. The color-coded marks under the sequence axis indicate the positions of important residues from Tables 3 and 4, see the legend at the bottom: ACT – Ca$^{2+}$-binding activation site, INH – M$^{2+}$-binding inhibition site. ATP – ATP-binding site, CFF – caffeine/xanthine binding site, S23 – S23 loop, S45 – S45 linker, GATE – channel gating residues.

the regulation of channel gating [37,47]. Individual model structures contained 1 – 4 branches in various proportions. One RyR1 and one RyR2 structure contained just a single activation branch.

**Interaction between the calcium activation and calcium inhibition networks.** The model RyR structures and their interacting pathways are correlated in Table 7 (the complete list of the critical residues is given in S3 Table). Each of the six intra-monomeric inhibition branches interacts with at least one activation branch. The largest number of interactions occurs in branch I1, which interacts with 4 activation branches in 2 RyR1-O, 1 RyR1-P, 2 RyR2-C, and 1 RyR2-O structures. The smallest number of interactions occurs in branches I2 and I3, which interact only with A2 and only in the RyR2-O structure 7ua9, which does not present other interactions. On the other hand, the activation branch A2 interacts with 3 inhibition branches in 1 RyR1-C, 1 RyR1-P, and 2 RyR2-O structures. Branches I7 – I9 and branch A9 presented no interactions in the allosteric network. A systematic interaction pattern for branches and structures was best revealed when the branches of the networks were connected graphically (Figs 8 and S2).

**Table 6. Fractional occurrence of branches in the activation network pathway.**

| Main branches connecting the ACT and GATE sites | | Fraction of paths (%) | | | | | | | | |
|---|---|---|---|---|---|---|---|---|---|---|
| | | RyR1 | | | | | RyR2 | | | |
| | | C | P | O | O† | I† | C | C† | O† | O |
| | | 7k0t | 7tzc | 7m6l | 7tdh | 7tdg | 7vmm | 7ua5 | 7ua9 | 7vmp |
| Name | ATP-site branches | | | | | | | | | |
| A1 | -CTD – U - ATP - S6- | | | 100 | | 20 | 100 | 80 | | 18 |
| A2 | -CTD – U - ATP - S6- | 60 | 38 | | | | | | 7 | 21 |
| A3 | -CTD – U - ATP - S6- | 40 | | | | | | | | |
| A4 | -CTD – U - ATP - S6- | | | | 69 | | | | | |
| A5 | -CTD - CFF - U - ATP site - S6- | | | | | | | | | 25 |
| A6 | -CTD – U - ATP - S45 - S6- | | | | | 77 | | 20 | | |
| | Subtotal | 60 | 38 | 100 | 69 | 97 | 100 | 100 | 7 | 64 |
| | S45 branches | | | | | | | | | |
| A7 | -CTD - U - S45 - S6- | | | | 14 | | | | | |
| A8 | -CTD - U - S45 - S6- | | | | | | | | 93 | |
| | Subtotal | 0 | 0 | 0 | 14 | 0 | 0 | 0 | 93 | 0 |
| | CFF-site branch | | | | | | | | | |
| A9 | -CTD - CFF– U - S6- | | 61 | | 17 | | | | | 36 |
| | Subtotal | | 61 | | 17 | | | | | 36 |
| | TOTAL | 100 | 99 | 100 | 100 | 97 | 100 | 100 | 100 | 100 |

ACT and GATE are defined in Table 4. The abbreviations CTD, U, ATP, CFF, S45, and S6 refer to the Central domain, U-motif, ATP-binding site, CFF-binding site, S45 linker, and S6 helix, respectively, which contribute by one or more residues to the path (see S2 Table for the list of critical residues of individual branches).

†Marks a pair of RyR1 and RyR2 structures obtained in the same experiment.

**Table 7. Interactions between pathways of the activation and inhibition network.**

| | A1 | A2 | A3 | A4 | A5 | A6 | A7 | A8 | A9 |
|---|---|---|---|---|---|---|---|---|---|
| I1 | RyR1-O 7m6l$ RyR2-C 7vmm$ RyR2-C 7ua5$ RyR2-O 7vmp$ | RyR1-P 7tzc RyR2-O 7vmp | | RyR1-O 7tdh | RyR2-O 7vmp | | | | |
| I2 | | RyR2-O 7ua9 | | | | | | | |
| I3 | | RyR2-O 7ua9 | | | | | | | |
| I4 | RyR1-I 7tdg | RyR1-P 7tzc | | | | RyR1-I 7tdg$ RyR2-C 7ua5$ | | | |
| I5 | | | | RyR1-O 7tdh | | | | | |
| I6 | | RyR1-C 7k0t | RyR1-C 7k0t | RyR1-O 7tdh | | | RyR1-O 7tdh$ | | |
| I7 | | | | | | | | | |
| I8 | | | | | | | | | |
| I9 | | | | | | | | | |
| I10 | | | | | | RyR1-I 7tdg# | | | |
| I11 | | | | | | RyR1-I 7tdg# | | | |
| I12 | | | | | | | | RyR2-O 7ua9$,# | |

$Marks structures where the same residue of the ATP-binding site or the S45 linker, participates in both the inhibition and activation network.

#Marks the interaction of the inter-monomeric inhibition pathway with the activation pathway of the anticlockwise-neighboring monomer.

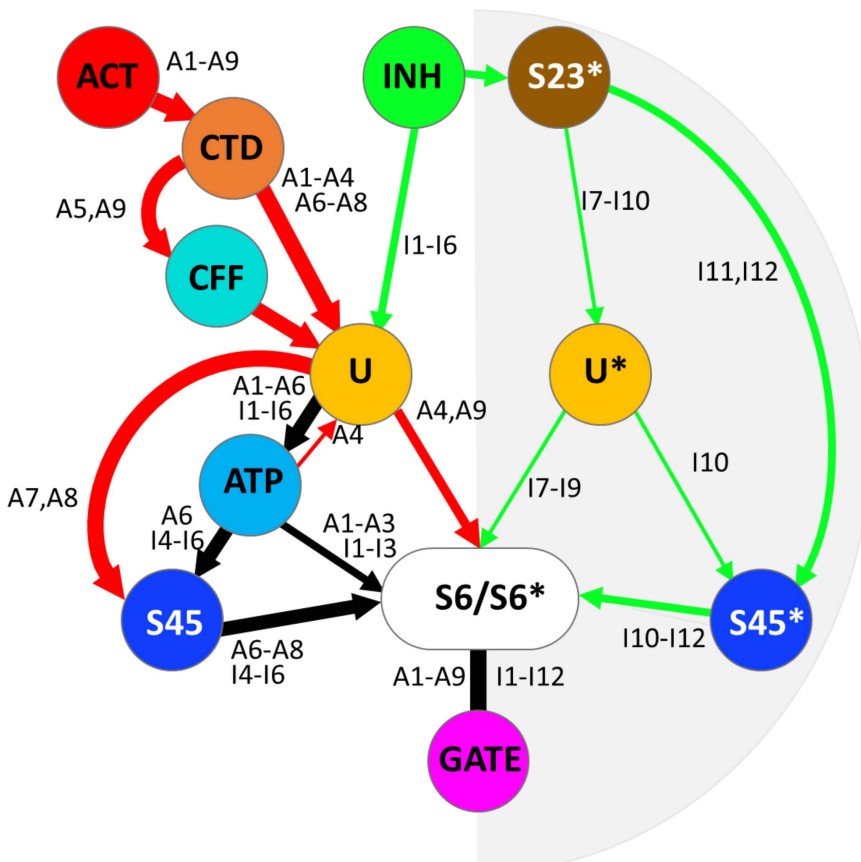

**Fig 8. Divalent ion-related allosteric networks of the ryanodine receptor.** The scheme integrates the intra-monomeric (white background) and the inter-monomeric pathways (gray background). Note that both pathway types are present in each monomer. Red – the activation network. Green – the inhibition network. Black – the shared activation and inhibition sub-paths. The branches are labeled in correspondence to Tabs 3–5 - 5. ACT – the activation site; INH – the inhibition site; CTD – the C-terminal domain; CFF – the caffeine binding site; S23 – the S23 loop; U, U* - the U-motifs; ATP – the ATP binding site; S45, S45* - the S45 linkers; S6/S6* - the S6 helices; GATE - the gating site. The spatial arrangement of the pathways of two example structures (the RyR1 inactivated structure 7tdg and the RyR2 open structure 7ua9) is shown in S2 Fig.

A search for important residues in the allosteric network revealed several residues shared by both networks. Each analyzed RyR structure involved U-motif residues I4218 and F4219 in either the inhibition allosteric network (7tdh, 7tzc, 7ua9), or the activation allosteric network (7tdg and 7k0t), or both (7m6l, 7vmm, 7ua5, 7vmp). In all RyR structures except RyR1-C 7k0t and RyR2-O 7ua9, at least one branch of the allosteric network contained the U-motif residues K4214 and W4205. In all RyR structures except RyR1-C 7k0t and RyR2-O 7ua9, the activation and inhibition branches converged either before or at these U-motif residues, or in the case of 7tdh at the CTD residue Q4947 that follows the U-motif residue F4219 in the pathway. Thus, the U-motif residues I4218, F4219, K4214, and W4205 emerge as essential for the allosteric interaction between the activation and inhibition pathways. The residues I4218 and K4214 also participate in the ATP binding site.

In all model structures, the majority of activation and inhibition pathways encompassed at least one ATP-binding residue, therefore these pathways may be speculated to affect the affinity of the ATP binding site and regulation of RyR activity by ATP.

It should be underlined that the intra-monomeric pathway of the inhibition network interacts with the activation network in all 9 investigated model structures. The U-motif appears central to mixing signals, incoming from the activation and inhibition sites and passing them through the ATP-binding site, and/or S45, or directly to the S6 segment. The S6 segment appears central to the signal integration to control the pore open/closed state in all pathways.

## The allosteric model of the RyR channel operation

The assumed *modus operandi* common to RyR channel isoforms across species should consider the single-channel activity as the result of structural transitions. More specifically, under the given experimental conditions the structural states and their allosteric interactions should have counterparts in the functional states and their allosteric transitions. A theoretical model of the ion-dependent RyR operation based on the results of structural analysis should approximate the single-channel data collected at various concentrations of divalent ions. The experimental structural and functional data indicate that:

(1) The gate structure of the S6 segment is the only structure that controls the permeation state of the channel pore in both RyR1 and RyR2. This means that both the activation and the inactivation processes act on the same gate. Consequently, the activation and the inactivation are inter-dependent processes.

(2) The ion-binding activation site is a self-contained structure composed of equivalent CD and CTD residues in both RyR1 and RyR2. It has a much higher affinity to $Ca^{2+}$ than $Mg^{2+}$ ions, and its allosteric coupling to the channel pore stabilizes the gate in the open state more effectively when occupied by $Ca^{2+}$ than any other ion.

(3) The ion-binding inhibition site is formed by the EF1 and EF2 ion-binding loops in both RyR isoforms. It has a similar low affinity to $Ca^{2+}$ and $Mg^{2+}$ ions, and its allosteric coupling to the channel pore stabilizes the gate in the closed position when occupied by either divalent ion. The binding-to-inactivation allosteric coupling efficiency is higher in RyR1 than in RyR2.

(4) RyR isoforms adopt mutually congruent structural and functional macrostates. The bound/unbound state of the ligand binding sites defines the state and the structure of the RyR molecule. The equilibrium between macrostates depends on the concentrations and binding constants of ligands/divalent ions.

(5) The closed, primed, and inactivated RyR structures have the gate of all 4 monomers in the closed position. The open RyR structure has the gate of all 4 monomers in the open position.

(6) The primed state is defined by activators bound to their respective ATP-binding and/or caffeine-binding sites.

(7) The ion-dependent allosteric network of RyR consists of the intra- and inter-monomeric pathways. The intra-monomeric activation and inhibition pathways merge at the U-motif near the ATP and CFF binding pockets. The inter-monomeric pathway is a specialized ion-dependent inhibitory path more prominent in RyR1.

(8) In the absence of a structure of the inactivated RyR2, the model assumes that such a structure exists at high concentrations of divalent ions and differs from the inactivated RyR1 structure by the extent of EF-hand - S23* coupling.

**Construction of the model of RyR operation.** The analysis of homotetrameric RyR structures revealed that all four RyR monomers in concord adopt one of four principal structural macrostates – the closed, the primed, the open, and the inactivated. The structure of macrostates depends on the concentrations of divalent ions, ATP, and caffeine. Closed and primed macrostates could be combined into a single closed macrostate of the model since both are closed and cannot be functionally distinguished at a constant ATP concentration. Therefore, we constructed the model of RyR operation, termed the COI (closed-open-inactivated) model, in which we assigned a functional macrostate corresponding to each of the closed/primed, open, and inactivated structural macrostates (Fig 9A). The three macrostates were interconnected so that each could be directly transformed into another since each state was observed at a range of calcium concentrations.

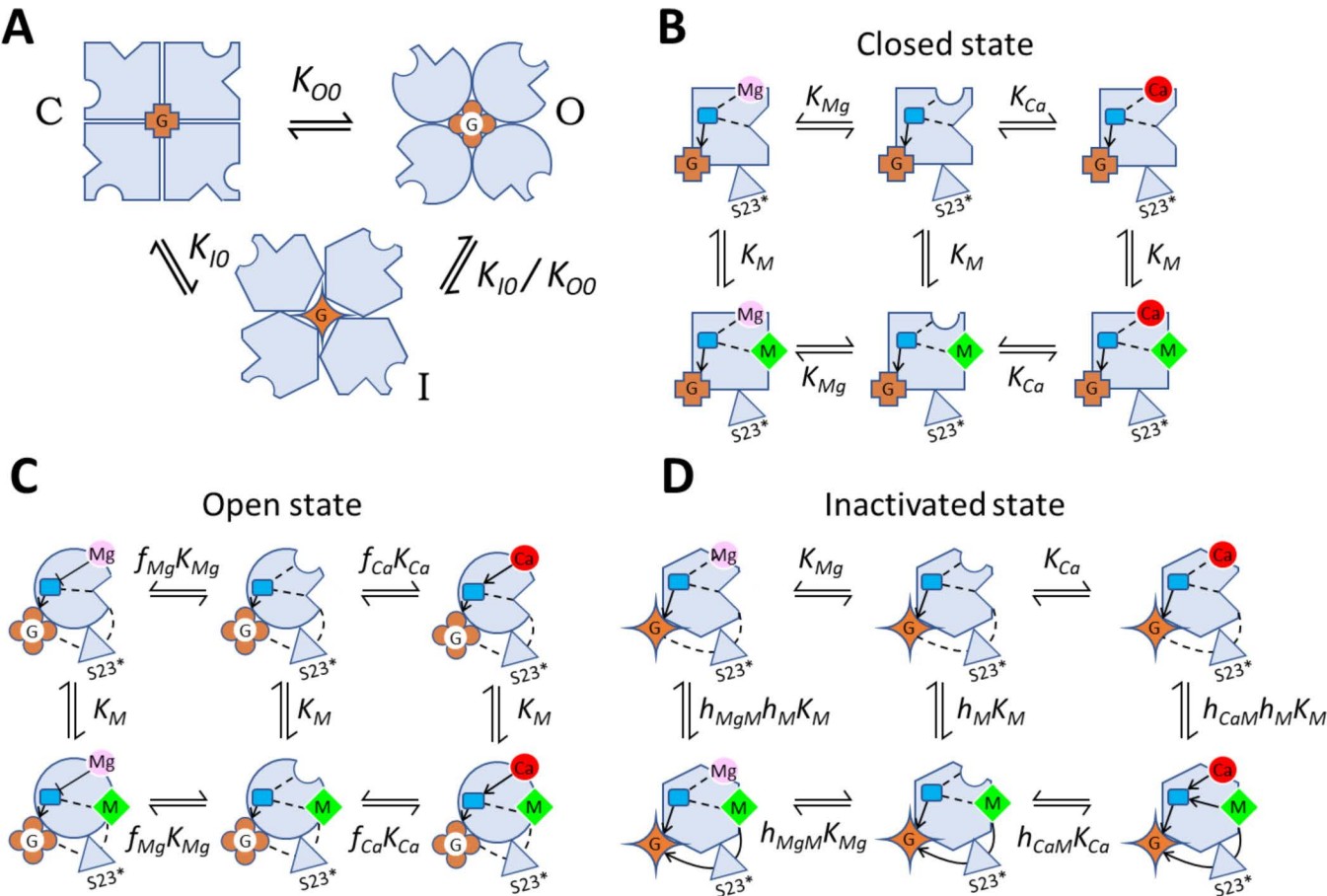

**Fig 9. The COI model of RyR channel operation. (A)** Transitions between the closed, open, and inactivated macrostates of ligand-free RyR. Equilibrium constants of transitions between the respective macrostates are indicated. **(B)** The ligand-binding transitions of a monomer in the closed macrostate. **(C)** The ligand-binding transitions of a monomer in the open macrostate. **(D)** The ligand-binding transitions of a monomer in the inactivated macrostate. (B – D) Gray shapes represent RyR monomers in the respective macrostates as depicted in A. Equilibrium constants of transitions between the respective states are indicated. Lines and arrows represent the allosteric interaction pathways. Dashed lines - no allosteric regulation. Blunt arrow (⊢) - negative allosteric interaction. The pointed arrows (←) - positive allosteric interaction. Round groove – the activation site. Pointed groove – the inhibition site. Blue square – the interaction site of the activation and the intra-monomeric inhibition pathways (ATP-binding site). Red circles – $Ca^{2+}$; pink circles – $Mg^{2+}$; green diamonds – $M^{2+}$ ($Ca^{2+}$ or $Mg^{2+}$). Brown shapes labeled G - the channel pore with the gate in the closed, open, or inactivated state. The grey triangle represents the S23 loop of a neighboring monomer.

As in our previous RyR gating models [3,30,59], transitions between channel states were considered spontaneous, with the unliganded RyR strongly preferring the closed state. A transition between macrostates transpires as a cooperative conformational change of all four monomers that occurs even without ligand binding, albeit with low probability. The binding of ligands changes the relative energies of macrostates, thus also the likelihood of finding the channel in a particular macrostate.

The operational schemes in Fig 9 describe transitions between (A) and within (B, C, D) individual macrostates respecting the identified allosteric pathways (A1-A9; I1-I12; Fig 8) as described by Eq. 3. Transitions between the unliganded macrostates: closed (squares), open (circles) and inactivated (hexagons) are illustrated in Panel A. Transitions between binding states of the closed macrostate are presented in Panel B. Since the closed macrostate is the reference state, transitions between binding states do not contain allosteric coefficients in this case. Transitions between binding states of

the open macrostate are shown in Panel C. The activation allosteric coefficients $f_{Ca}$ and $f_{Mg}$ specify the effect of divalent ion binding to the $Ca^{2+}$-activation site on the coupling efficiency of the activation pathway (A1-A9). Transitions between binding states of the inactivated macrostate are displayed in Panel D. The interaction factors $h_{CaM}$ and $h_{MgM}$ quantify the interaction between the activation site and the inhibition site resulting from the existence of common sub-paths in the intra-monomeric network (A1-A6; I1-I6). The classical state diagram of the COI model is explained in S3 Fig.

In the equilibrium, the probability of occurrence of a macrostate is given by the respective equilibrium constants and allosteric factors. Allosteric factors multiply equilibrium constants to reflect the changes in the energy of a macrostate relative to the closed macrostate, caused by the binding of an ion. If the affinity of an ion to the new macrostate does not change, the allosteric factor is 1, if the affinity increases the factor is <1; and if the affinity decreases the factor is >1. Transitions between states were characterized in the COI model by allosteric factors of two types – allosteric coefficients and interaction factors. The allosteric coefficients define the binding affinity of a single ligand to the O and I macrostates relative to the ligand's affinity in the closed macrostate. The allosteric coefficients of $Ca^{2+}$ and $Mg^{2+}$ binding at the activation site are defined as $f_{Ca}$ and $f_{Mg}$ in the open macrostate, and $h_{Ca}$ and $h_{Mg}$ in the inactivated macrostate. The inhibition site does not discriminate between $Ca^{2+}$ and $Mg^{2+}$, therefore the allosteric coefficient is defined as $f_M$ in the open macrostate and $h_M$ in the inhibited macrostate. The interaction factors reflect the changes of ion affinity induced by the simultaneous occupation of both the activation and inhibition sites and are defined as $g_{CaM}$ and $g_{MgM}$ in the closed macrostate, $f_{CaM}$ and $f_{MgM}$ in the open macrostate, and $h_{CaM}$ and $h_{MgM}$ in the inactivated macrostate.

Since both the activation pathway and the intra-monomeric inhibition pathway lead through the binding site for ATP, the values of $f_{Ca}$, $f_{Mg}$, $h_{CaM}$, and $h_{MgM}$ should depend on the binding of ATP; however, since only the absence of ATP or a single concentration of ATP was considered here, its effect was convolved into the respective allosteric factors, which were allowed to depend on the presence of ATP. To reduce the number of parameters (see above), the interaction between the activation and inhibition binding sites in the COI model was constrained only to the inhibited macrostate, as depicted in Fig 9D.

The inactivation allosteric coefficient $h_M$ specifies the impact of divalent ion binding to the RyR inhibition site on the gating site, transmitted through the S23* loop of the neighbor monomer. Since the inter-monomeric inhibition pathway does not involve any ATP-binding residues, the value of $h_M$ is independent of the presence of ATP at either monomer.

## Derivation of the model open probability equation

In the COI model, the equilibrium probability of the open state, $P_O$, can be described by Eq. 1, using the approach of statistical mechanics [49]:

$$P_O = \frac{\frac{1}{K_{O0}} w_O^4}{\frac{1}{K_{O0}} w_O^4 + w_C^4 + \frac{1}{K_{I0}} w_I^4} \tag{1}$$

where $w_O$, $w_C$, and $w_I$ are the statistical weights of the open, closed, and inactivated macrostates, respectively; the exponent 4 reflects the cooperative action of 4 monomers, and $K_{O0}$ and $K_{I0}$ are the equilibrium constants of the open and the inhibited ligand-free macrostate, respectively.

Changes in RyR macrostates induced by the binding of ions are expressed through the dependences of $w_O$, $w_C$, and $w_I$ on the ion-binding equilibria. The weight factors of the resulting COI model, calculated according to Marzen et al. [49] and Horrigan and Aldrich [60], are then given by:

$$w_C = 1 + \frac{[Ca]}{K_{Ca}} + \frac{[Mg]}{K_{Mg}} + \frac{[M]}{K_M} + \frac{[Ca][M]}{g_{CaM}K_{Ca}K_M} + \frac{[Mg][M]}{g_{MgM}K_{Mg}K_M} \tag{2A}$$

$$w_O = 1 + \frac{[Ca]}{f_{Ca}K_{Ca}} + \frac{[Mg]}{f_{Mg}K_{Mg}} + \frac{[M]}{f_M K_M} + \frac{[Ca][M]}{f_{CaM}f_{Ca}K_{Ca}K_M} + \frac{[Mg][M]}{f_{MgM}f_{Mg}K_{Mg}K_M} \tag{2B}$$

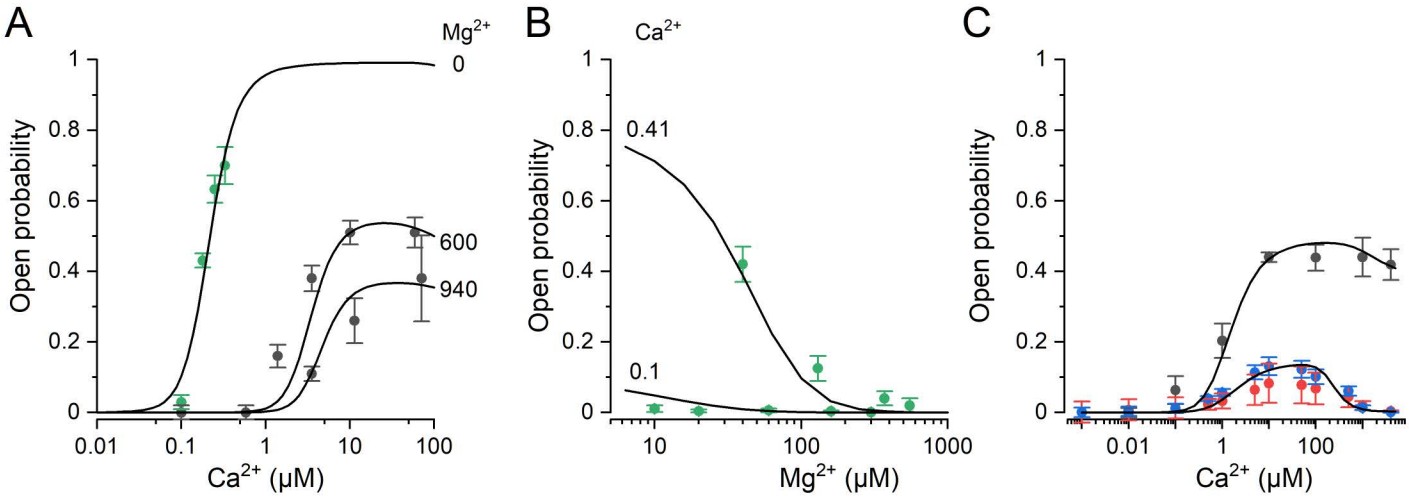

**Fig 10. Description of RyR open probability data by the COI model of RyR operation.** The data points represent the mean and SEM of the single-channel open probability values read from the referenced publications (see below). The curves represent the theoretical predictions of the COI model (Eq. 3) with the corresponding parameters for individual data groups in Table 8. **A** – Calcium dependence of the single-channel $P_O$ at cytosolic $[Mg^{2+}]$ of 0, 600, and 940 µM (indicated next to the data) in the presence of 3 mM cytosolic ATP and 1 mM luminal $Ca^{2+}$. Black points – canine RyR2 channels [36]. Green points - rat RyR2 channels [61]. Lines - theoretical predictions of the COI model with parameters for the "RyR2 + ATP" group in Table 8. **B** – Magnesium dependence of the single-channel $P_O$ at cytosolic $[Ca^{2+}]$ of 0.1 and 0.41 µM (indicated next to the data) in the presence of 3 mM cytosolic ATP and 1 mM luminal $Ca^{2+}$. Green points - rat RyR2 channels [61,62]. Lines - theoretical predictions of the COI model with parameters for the "RyR2 + ATP" group in Table 8. **C** – Calcium dependence of the single-channel open probability obtained in the absence of cytosolic ATP and $Mg^{2+}$ and with 10 µM luminal $Ca^{2+}$ [26,63]. Black points - canine RyR2 channels [26]. Blue points - human RyR1 channels [63]. Red points – rabbit RyR1 channels [26]. Lines - theoretical predictions of the COI model with parameters for the "RyR1 − ATP" or "RyR2 − ATP" groups in Table 8.

$$w_I = 1 + \frac{[Ca]}{h_{Ca}K_{Ca}} + \frac{[Mg]}{h_{Mg}K_{Mg}} + \frac{[M]}{h_M K_M} + \frac{[Ca][M]}{h_{CaM}h_M K_{Ca}K_M} + \frac{[Mg][M]}{h_{MgM}h_M K_{Mg}K_M}$$

(2C)

where the square brackets stay for the concentration of an ion in mol.l⁻¹, and [M] = [Ca] + [Mg] stays for the concentration of divalent ions M. The individual terms at the right side of the equations characterize individual ion-binding state transitions of a monomer in the closed, open, and inactivated macrostates.

Equation 1 contains two constants and three independent variables, which are described in Eq. 2 by three variables (two independent concentrations of $Ca^{2+}$ and $Mg^{2+}$, and their sum $M^{2+}$) and 15 parameters characterizing the chemical equilibria. The single-channel data in the literature consist of concentration dependencies of open probability under different experimental conditions that do not always cover the full range and often display relatively large errors of measurement. This data paucity together with the relatively simple and similar shapes of ion-dependencies (see Fig 10) does not allow direct determination of all 17 parameters per concentration dependence by fitting Eqs. 1 and 2 to data points due to the parameter redundancy problem. Since solving this problem experimentally is not practical or even feasible, we resolved it by a substantial reduction of the number of parameters in Eq. 2A–C, while respecting the observed allosteric pathways. We used the following practical assumptions: The allosteric effect of $Ca^{2+}$ and $Mg^{2+}$ bound at the activation site occurs only in the open macrostate, meaning that the values of $h_{Ca}$ and $h_{Mg}$ in Eq. 2C can be set to 1, i.e., the allosteric effect of ion binding to the activation site on the inactivated macrostate can be considered negligible. Similarly, the allosteric effect of $M^{2+}$ bound at the inhibition site occurs only in the inactivated macrostate, i.e., $f_M$ in Eq. 2B can be set to 1. Finally, the interaction between the activation and the inhibition site occurs only in the inactivated macrostate, i.e., the values of interaction factors $f_{CaM}$, $g_{CaM}$, $f_{MgM}$, and $g_{MgM}$ in Eq. 2A and B can be set to 1. By applying these suppositions, the Eq. 2A–C can be simplified to Eq. 3A–C that contain only 8 independent parameters:

$$w_C = 1 + \frac{[Ca]}{K_{Ca}} + \frac{[Mg]}{K_{Mg}} + \frac{[M]}{K_M} + \frac{[Ca][M]}{K_{Ca}K_M} + \frac{[Mg][M]}{K_{Mg}K_M} \tag{3A}$$

$$w_O = 1 + \frac{[Ca]}{f_{Ca}K_{Ca}} + \frac{[Mg]}{f_{Mg}K_{Mg}} + \frac{[M]}{K_M} + \frac{[Ca][M]}{f_{Ca}K_{Ca}K_M} + \frac{[Mg][M]}{f_{Mg}K_{Mg}K_M} \tag{3B}$$

$$w_I = 1 + \frac{[Ca]}{K_{Ca}} + \frac{[Mg]}{K_{Mg}} + \frac{[M]}{h_M K_M} + \frac{[Ca][M]}{h_{CaM}h_M K_{Ca}K_M} + \frac{[Mg][M]}{h_{MgM}h_M K_{Mg}K_M} \tag{3C}$$

i.e., together with Eq. 1, we remained with only 10 independent parameters (see the list in Table 8).

### Description of RyR operation by the COI model

The published single-channel data on $Ca^{2+}$ and $Mg^{2+}$ dependence of the open probability of RyR1 and RyR2 are quite limited by the range, completeness, and compatibility of experimental conditions. The experimental data for validation of the model were taken from experiments of Fill et al. [63], Chu et al. [26], Zahradnikova et al. [36], Tencerova et al. [61], and Li et al. [62] that conformed with conditions used for collecting RyR molecules for cryo-electron microscopy determination of their structure. These include isolated cardiac microsomal vesicles incorporated into lipid bilayers either under near-physiological conditions with a total ATP concentration of 2–3 mM, free $Mg^{2+}$ in the range of 0 - 1.3 mM, and free $Ca^{2+}$ in the range of 0.1 – 100 µM at the cytosolic side and 0.1 - 1 mM at the luminal side (rat or canine cardiac microsomes [36,61,62]) or under simpler conditions in the absence of $Mg^{2+}$ and ATP, at free cytosolic $Ca^{2+}$ in the range of 0.1 – 4000 µM and approximately 10 µM luminal $Ca^{2+}$ (canine cardiac microsomes and rabbit or human skeletal muscle microsomes [26,63]).

The whole experimental dataset consists of 8 functional relations with 54 data points represented by the mean and the standard error of the mean (Fig 10), obtained under five different experimental conditions that did not cover the full range and often displayed relatively large errors of measurement. If fitted separately, they would need 64 separate parameter values. Therefore, to reduce the number of parameter values, we combined published data into three groups by neglecting the differences in RyR single-channel activity between species (Table 8, and Fig 10A–C). We combined data obtained in rat [61,62] and canine [36] RyR2 channels in the presence of ATP, luminal $Ca^{2+}$, and at various cytosolic $Ca^{2+}$ and $Mg^{2+}$ levels, to estimate the parameter values of RyR2 transitions at near-physiological conditions (Table 8, "RyR2 + ATP" group). The data from canine RyR2 [26] in the absence of $Mg^{2+}$ and ATP and at a low level of luminal $Ca^{2+}$ formed the group "RyR2 − ATP" (Table 8). We combined human [63] and rabbit [26] RyR1 in the absence of $Mg^{2+}$ and ATP and at a low level of luminal $Ca^{2+}$ (Table 8, "RyR1 − ATP" group). The groups "RyR1 − ATP" and "RyR2 − ATP" allowed us to characterize the difference between RyR1 and RyR2 channels. The groups "RyR2 + ATP" and "RyR2 − ATP" allowed us to characterize the effect of ATP on RyR2. This procedure led to the reduction of the number of parameter values to 24 for the whole system of equations approximating the three data groups (Table 8).

The algorithm used for fitting (see Methods) provided detailed information on the quality of fit, including thorough control of the fitting process and a detailed description of the fitting results, namely, the standard error of the fit, parameter dependency, and statistical significance of the estimated parameter value. This allowed us to solve the anticipated problems of convergence and over-parametrization by fixation of parameter values with the highest dependency and sharing parameter values between groups when they were not significantly different from each other.

In the first step, we fixed several parameters at previously published values or considered an *a priori* estimate (see Table 8). The value of $K_{O0}$ was set at the value estimated in our previous study based on a principally similar allosteric gating model of RyR2, where its determination was enabled by the existence of data for the open probability of RyR channels with a variable number of calcium activation-deficient monomers [3]. The $K_{O0}$ value was set the same for all three data

**Table 8. The best-fit parameter values of the COI model (**Equation 3**).**

| Parameter | Description | RyR2 + ATP | RyR2 − ATP | RyR1 − ATP | Reference or justification |
|---|---|---|---|---|---|
| $K_{O0}$ | Equilibrium constant for the transition C ↔ O (Eq. 1) | **10 800** | | | Taken from [3] |
| $K_{I0}$ | Equilibrium constant for the transition C ↔ I (Eq. 1) | **10 800** | | | Set equal to $K_{O0}$ |
| $K_{Ca}$ (µM) | Dissociation constant of $Ca^{2+}$ from the activation site (Eq. 3) | 0.59 ± 0.15 | | | Best fit value |
| $K_{Mg}$ (µM) | Dissociation constant of $Mg^{2+}$ from the activation site (Eq. 3) | 36.3 ± 4.3 | | | Best fit value |
| $K_M$ (µM) | Dissociation constant of $Mg^{2+}$ from the inhibition site (Eq. 3) | **546** | | | Taken from [30] |
| $f_{Mg}$ | Allosteric factor of $Mg^{2+}$ at the activation site (Eq. 3) | **3.25** | | | Taken from [30] |
| $f_{Ca}$ | Allosteric factor of $Ca^{2+}$ at the activation site (Eq. 3) | 0.029 ± 0.005 | 0.10 ± 0.003 | 0.15 ± 0.01 | Best fit values |
| $h_M$ | Allosteric factor of $M^{2+}$ at the inhibition site (Eq. 3) | **1** | **1** | 0.28 ± 0.10 | See text |
| $h_{CaM}$ | Interaction factor between $Ca^{2+}$ at the activation site and $M^{2+}$ at the inhibition site (Eq. 3) | 0.016 ± 0.003 | 0.11 ± 0.04 | | Best fit value |
| $h_{MgM}$ | Interaction factor between $Mg^{2+}$ at the activation site and $M^{2+}$ at the inhibition site (Eq. 3) | **1** | **1** | **1** | See text |
| $h_M \cdot h_{CaM}$ | | *0.016 ± 0.003* | *0.11 ± 0.04* | *0.031 ± 0.11* | *assuming normal distribution* |
| COD | Coefficient of determination | 0.953 | | | |

Values in bold were fixed in all calculations. Values presented as the mean ± error represent the best-fit values and the standard error of the fit. Parameters in the column "RyR2 + ATP" correspond to fits to the black and green data points in Fig 10A and 10B. Parameters in columns "RyR2 − ATP" correspond to fits to the black data points in Fig 10C. Parameters in columns and "RyR1 − ATP" correspond to fits to the blue and red data points in Fig 10C.

groups, based on the structural similarity of RyR1 and RyR2 in the closed state. The stability constant of the unoccupied inactivated state ($K_{I0}$) was *a priori* considered equal to that of the unoccupied open state ($K_{O0}$) since we consider both the open and the inactivated macrostate in the absence of ligands thermodynamically equal.

The values of $f_{Mg}$ and $K_M$ were set in all three groups at their values estimated in our previous study on RyR2 channels based on fitting of the open probability, open times, and the rate of activation by a kinetic model of allosteric gating, see [30]. The equivalence of $f_{Mg}$ in RyR1 and RyR2 was based on the identical structure of the activation site in both isoforms. The equivalence of $K_M$ in RyR1 and RyR2 was based on the similar values of the ion-binding parameters of the EF-hands in RyR1 and RyR2 estimated in the section "The propensity of EF1 and EF2 loops to bind divalent ions". The ion dissociation constants $K_{Ca}$ and $K_{Mg}$ were shared between all data groups since the structure of the activation site in both isoforms is identical.

The parameter defining the strength of allosteric interaction between the $Mg^{2+}$ bound at the activation binding site and $M^{2+}$ bound at the inhibition binding site ($h_{MgM}$) could not be optimized for any isoform, due to the absence of Po data in the presence of $Mg^{2+}$ for RyR1 channels. The test of $h_{MgM}$ variation for RyR2 between 1 and 0.2 yielded a negligible impact on $\chi^2$ (5%) and coefficient of determination (0.3%) and did not affect significantly the remaining parameter values. Therefore, $h_{MgM}$ was set to 1 for all three groups. This solution practically disconnected $Mg^{2+}$ binding to the activation site from the intra-monomeric inhibition pathway.

The value of parameter $h_M$, which defines the strength of the allosteric effect of $M^{2+}$ bound at the inhibition site on the inter-monomeric pathway, was set to 1 for RyR2 data (groups "RyR2 + ATP" and "RyR2 − ATP"), in correspondence to the small intensity of the inter-monomeric pathway in the RyR2 isoform (Table 5). The effect of this simplification was tested by varying $h_M$ of RyR2 in the interval $0.25 \leq h_M \leq 1$, which returned negligible impact on $\chi^2$ (4%) and coefficient of determination (0.2%), a proportional change in $h_M$ for RyR1 data, a reciprocal change in $h_{CaM}$ of both RyR1 and RyR2, and undetectable change in the remaining parameter values. This interdependence means that the dataset does not allow the determination of the unequivocal values of $h_M$ and $h_{CaM}$ but only of their product $h_M \cdot h_{CaM}$ (Table 8, bottom line).

The parameters $h_{CaM}$ for the data groups "RyR2 − ATP" and "RyR1 − ATP", if optimized as independent parameters, were not significantly different from each other but the parameter values of $h_M$ and $h_{CaM}$ were highly correlated and their standard errors of the fit were large; therefore, the parameter $h_{CaM}$ was shared for both "RyR2 − ATP" and "RyR1 − ATP" data groups in the final optimization. In the end, eight parameters of the COI model were set free for fitting the experimental data, as summarized in Table 8. It should be noted that the minimization criterion was the minimal chi-square value for the whole dataset.

The solutions of the COI model for the dependences of the RyR1 and RyR2 open probabilities on concentrations of $Ca^{2+}$ and $Mg^{2+}$ ions (Fig 10, lines; Table 8, parameter values) indicate that the model complies satisfactorily with the data obtained at a wide range of conditions. These include four species, two RyR isoforms, and the presence or absence of cytosolic ATP, $Mg^{2+}$, and luminal $Ca^{2+}$ ions.

It should be noted that only $f_{Ca}$ and $h_M$ were allowed to vary with the isoform of RyR, while $f_{Ca}$ and $h_{CaM}$ were allowed to vary with the presence of ATP, see their best-fit values in Table 8. All equilibrium constants ($K_{O0}$, $K_{Ca}$, $K_{Mg}$, $K_{I0}$, $K_M$) had common values for the whole dataset (Table 8).

As mentioned above, the allosteric inhibition factor $h_M$ was fixed to 1 for fitting RyR2 data; this means that the inter-monomeric inhibition pathway was made independent of the occupation of the inhibition site by $M^{2+}$. Still, however, the intra-monomeric inhibition pathway is operational and RyR2 inactivation is thus controlled by the interaction factor $h_{CaM}$, which affects the open probability when the activation site is occupied by $Ca^{2+}$ and the inhibition site is occupied by $Ca^{2+}$ or $Mg^{2+}$ (Fig 9D). In other words, the simultaneous occupation of both the activation and the inhibition site by divalent ions makes the inhibited macrostate energetically more favorable also in RyR2. Therefore, due to the interaction of the intra-monomeric inhibition pathway with the activation pathway, the reduction of $P_O$ can occur even without the recruitment of the inter-monomeric pathway; the strength of this interaction, $h_{CaM}$, determines the sensitivity to inhibition by divalent ions in both RyR isomers.

The analysis of inactivation parameters of RyR1 and RyR2 (Table 8) shows that in the absence of ATP, the overall inactivation (proportional to $1/(h_M.h_{CaM})$) is 3.5× more effective in RyR1 than in RyR2 (0.11/0.031 = 3.5). In the presence of ATP and luminal $Ca^{2+}$, the inactivation of RyR2 is about 7× more effective than in the absence of ATP (0.11/0.016 = 6.9). The relative efficacy of inactivation in RyR1 and RyR2 as well as in the absence and presence of ATP were not affected by the $h_M$ value set for RyR2. However, this value strongly affected the predicted efficacy of the inter-monomeric inhibition pathway relative to the intra-monomeric one. The relative importance of the inter-monomeric pathway increased with inverse proportion to the square of the set value of $h_M$.

In the COI model, the $Ca^{2+}$ and $Mg^{2+}$ dissociation constants of the activation site ($K_{Ca}$, $K_{Mg}$, Table 8) as well as the $M^{2+}$ dissociation constant of the inhibition site ($K_M$, Table 8) are the same for RyR1 and RyR2 independent of the species. Independent of the species is also the allosteric factor $f_{Ca}$ controlling the maximum open probability and the apparent sensitivity to activation by $Ca^{2+}$; however, it is significantly different in RyR1 and RyR2 as well as in the absence and presence of ATP in the case of RyR2 (Table 8). Moreover, the interaction factor $h_{CaM}$ is independent of the species and the RyR isoform in the absence of ATP, and it is the same for RyR2 in both species in the presence of ATP. In this sense, the COI model unifies the published single-channel data on RyR activity, which appeared inconsistent among isoforms, species, and experimental conditions.

## Discussion

To elucidate the mechanism of RyR operation, we compared the structures and the activity profiles of RyR1 and RyR2 channels at a range of cytosolic $Ca^{2+}$ and $Mg^{2+}$ concentrations. Examination of the EF-hand region as a candidate divalent ion-binding inhibition site in RyR1 and RyR2 structures indicated that in both isoforms the EF1 and EF2 have a similar predisposition to bind $M^{2+}$; thus the difference between RyR isoforms in the propensity to inactivation by divalent ions should reside in the transmissibility of the $M^{2+}$ binding signal from the inhibition site to the channel gate. Structural analysis

of 26 RyR1 and 17 RyR2 structures revealed a single major difference in the configuration of RyR1 and RyR2 domains that resided in the orientation of the EF-hand relative to the S23 loop of a neighbor monomer (Figs 4 and 5). Other domains in the core RyR structures differed in their positions in the closed state relative to that in the open, primed, or inactivated states; nevertheless, they were similar in RyR1 and RyR2. These facts allow us to conclude that the EF1 and EF2 loops represent the inhibitory $M^{2+}$ binding site in both RyR isoforms since they bind divalent ions well, and the interaction between the EF-hand region and the S23 loop is stronger in RyR1 than in RyR2. Thus, the difference in the inactivation of RyR1 and RyR2 by $M^{2+}$ appears to be explainable simply by the different electrostatic interactions of their EF-hands with the S23* loops specific for RyR isoforms.

The situation turned even more interesting in the analysis of the allosteric network between the divalent ion-binding inhibition site and the gate and between the $Ca^{2+}$-binding activation site and the gate. This revealed two major pathways in the core structure of both RyR isoforms. The robust intra-monomeric pathway that shares the inhibition and the activation branches and involves residues of the U-motif near and including the ATP- and caffeine-binding sites was present in each of the closed, open, primed, and inactivated structural states. The less robust inter-monomeric pathway, present in all open, primed, and inactivated structural states of RyR1 and one open state of RyR2, uses the connection between the EF-hand region and the S23 loop of neighbor monomers. Both pathways may spread the inhibition signal to the common channel gate at the S6 helix either directly from the U-motif or via the S45 linker (Fig 8).

The question of how this complex structure may fulfill the RyR function was approached by mathematical modeling. We built an operational model based on structural data and applied it to the set of published single-channel data on the RyR channel open probability, measured under similar near-physiological conditions as the analyzed RyR structures. The operation model was built on the premise of the correspondence between structural and functional macrostates. We postulated the COI model with the closed, open, and inactivated, mutually connected macrostates, and with relative occupancy of the macrostates controlled by ion-binding equilibria among them. The primed state was not resolved from the closed macrostate. Each macrostate has the activation and inhibition divalent ion binding sites and reaction paths obeying the major allosteric pathways. The probability of the COI model reaching the open macrostate depends on the concentration of divalent ions, their respective binding constants, and four allosteric coefficients that regulate the transitions between the channel states. The values of model parameters were found by approximating the experimental dataset of 54 open probabilities from four laboratories, determined at a broad range of $Ca^{2+}$ and $Mg^{2+}$ concentrations, in RyRs isolated from skeletal and cardiac muscles of various mammals. We considered the interspecies differences negligible for simplification. This allowed us to combine the data into three groups with the number of data points allowing the approximating algorithms to converge to acceptable solutions. After the optimization of the parameter set, the COI model simulation provided reasonable correspondence to experimental data points.

The basic features of the COI model, common to RyR1 and RyR2, are the activation site with a high affinity for $Ca^{2+}$ and a low affinity for $Mg^{2+}$ and the inhibition site with a low affinity for both $Ca^{2+}$ and $Mg^{2+}$. The difference is in the values of some allosteric parameters that were dependent on either the RyR isoform or the presence/absence of ATP. The allosteric strength of $Ca^{2+}$ binding to the activation site for opening the gate was higher in RyR2 than in RyR1; the allosteric strength for closing the gate by $M^{2+}$ binding to the inhibition site was higher in RyR1 than in RyR2; but the interaction between the inhibition and activation sites was identical in RyR1 and RyR2. The presence of ATP and luminal calcium transpires as the increase in both the activation strength of $Ca^{2+}$ and the interaction strength between the inhibition and the activation site.

## Comparison with previous studies

This is the first study that addresses structural relationships related to divalent ion control of RyRs using a large set of structures. Previously, many authors made illuminating discoveries based primarily on a limited set of structures [6,64–66] or using one structure for each RyR state [22,58]. Our approach based on the analysis of a collection of compatible structures allowed generalization of the partial results on the RyR structure and revealed the interaction of allosteric pathways.

The importance of the U-motif residues I4218 and F4219, which we observed in the allosteric networks of most RyR1 and RyR2 model structures is underlined by the finding that the corresponding RyR2 residues I4172 and F4173 were found important for the interaction of the U-motif with the C-terminal domain and S6 helix, respectively [66], and their alanine substitution led to the gain of function for RyR2 [47].

Greene et al. [67] observed in molecular dynamics simulations that the S45 linker of all monomers has to assume the "open" position for the channel to be open, suggesting their intimate involvement with the transitions between macrostates. The residues T4825, I4826, S4828, and S4829, present in several branches of the activation and inhibition networks, were shown to be involved in RyR1 gating [47], and the mutation T4825I was shown to weaken RyR1 inhibition by $Ca^{2+}$ [37]. The existence of the inter-monomeric inhibition pathway in the open RyR2 structure 7ua9 and the presence of the functionally important S45 residues corresponding to rRyR1 T4825, I4826, S4828, and S4829 in its allosteric network suggests that the RyR2 channels might potentially be inactivated also through the inter-monomeric pathway. This is in agreement with the satisfactory fit of RyR2 data with $h_M$ reduced to 0.25. Uehara et al. [68] found that K4821 of S45 is a critical residue at which cytosolic and luminal inputs converge. We have observed this residue both in the activation and inhibition allosteric pathways of the inhibited RyR1 model structure 7tdg and the closed RyR2 structure 7ua5 and the inhibition pathways of three other RyR1 model structures (S1 and S2 Tables). The determination of the role of luminal $Ca^{2+}$ in RyR gating deserves further study, considering the crucial role of the highly dynamic $Ca^{2+}$ content of SR cisternae [69,70].

To date, allosteric pathways in ryanodine receptors have been examined only in one study [58]. These authors analyzed activation pathways in the closed 5t15, primed 5tap, and open 5tal RyR1 structures [5] and identified allosteric pathways common for the Ca-, ATP-, and caffeine-binding sites. Our results can be compared to those of Chirasani et al. [58] only partially since we have not included the structures analyzed by Chirasani et al. [58] in our analysis due to their lower resolution than the later structures 7m6l and 7tzc [7,71]. In the RyR1-C structure 5t15 [5], Chirasani et al. [58] identified in the Ca-activation network several residues that we identified in the RyR1-C structure 7k0t [22]: the CTD residue Y4994, the U-motif residue F4217, and the S6 segment residues Q4946 and E4942. The ATP-binding residue I4218 present in the activation pathway of 7k0t occurred only in a less important pathway of 5t15. The incomplete residue conformity in allosteric pathways may be caused by the occupation of the ATP-binding site by ACP in the structure 7k0t, and the empty ATP-binding site in 5t15. In the case of the open RyR1 structures, Chirasani et al. [58] observed in the Ca activation pathway of 5tal [5] the important ATP-binding residues K4214 and I4218, the CTD residues K4998 and Y4994, and the S6 segment residues Q4946, E4942, and D4938 leading to the gate as we observed in the structure 7m6l [71]. The subtle differences between our and the Chirasani et al. [58] study might be either due to the differences in the experimental conditions (concentrations of TCEP, ACP vs. ATP, ionic strength) or the limited resolution of the older structures [5].

Chirasani et al. [39] reported a significant increase of IC50 for $^3$H-ryanodine binding (RyR inhibition) by $Ca^{2+}$ (from ≈0.8 to ≈2.5 mM) and $Mg^{2+}$ (from ≈1.2 to ≈4.5 mM) upon introducing the mutation K4101D into the EF-hand region of RyR1. Mutations K4101A and K4101M also had a significant but smaller effect. On the other hand, mutations of the hydrogen-bonding partner of K4101 in the S23* loop to D4730K or D4730N did not affect RyR inactivation. In the light of our allosteric pathway analysis, these results can be explained by the transmission of the allosteric signal from K4101 not only to D4730 but also to I4731 (Fig 5B and S1 Table). Thus, the mutation of only D4730 is not sufficient to block the allosteric transmission through these domains. Interestingly, however, the K4101E mutation had an unexpected effect: in addition to increased IC50 it also increased the EC50 for $Ca^{2+}$-dependent activation of $^3$H-ryanodine binding [39], difficult to explain by current data.

Based on their structural analysis of the open RyR1 structure 5tal, Chirasani et al. (2024) [39] proposed that the narrow gap between the EF-hand domain and S23* loop of RyR1, necessary for H-bonding interactions between these two domains, is a consequence of the binding of $Ca^{2+}$ to the EF-hands. However, our structural and bioinformatics analysis (Figs 2 and 3) revealed a similar potential for interaction between the EF-hand region and the S23* loop not only in the inactivated but also in the open and primed RyR1 structures at intermediate calcium concentrations. The presence of

EF-hand domain – S23* hydrogen bonds in all RyR1 structures suggests that the proximity of the EF-hand domain and S23* loop is a structural trait distinguishing RyR1 from RyR2, not a consequence of $Ca^{2+}$ binding to the EF-hand.

The recent RyR1 structure 7umz [23] demonstrated the binding of $Mg^{2+}$ ion in the activation site, confirming thus the functional studies that established competition between $Ca^{2+}$ and $Mg^{2+}$ at the RyR activation site [35,36,59]. This structure, although obtained in the presence of 10 mM $Mg^{2+}$, was defined as the closed state, since the structure of the activation site resembled that in the absence of bound $Ca^{2+}$ despite the presence of a bound $Mg^{2+}$ ion [23]. However, the low flexion angle (-2.8°) and the configuration of the EF-hand region relative to the S23* loop reported by Nayak et al. [23], as well as their distance (Fig 2, magenta) and the number of interactions (Fig 3, magenta) compared in this work, resemble more the primed and the inactivated state than the closed state. However, the position of the EF-hand region relative to the Central domain differs from RyR1 in both the closed and the $Ca^{2+}$-inactivated states (Fig 4, magenta), similarly as in RyR2 structures. This peculiar conformation can be speculated to result from the independence of the functional macrostate (closed/ primed – open – inactivated) from the occupation status of the monomer binding sites, as postulated in the MWC theorem [48] and embedded in the COI model. The parameters of the COI model for RyR1 channels in the absence of ATP (Table 8) predict the channel to reside predominantly in the closed state. However, variation of $h_{MgM}$ in the interval between 1 and 0.2, which was shown to be acceptable for RyR2 channels, results in a progressive shift towards the inactivated state (from <1% at $h_{MgM}$ = 1 to ≈80% at $h_{MgM}$ = 0.2). Thus, distinguishing the state of the channel at 10 mm $Mg^{2+}$ would require RyR1 open probability data at a large interval of $Ca^{2+}$, $Mg^{2+}$, and ATP concentrations, not presently available.

## Implications

The structural analysis presented here explains the paradox of different sensitivity of RyR1 and RyR2 to inactivation by $Ca^{2+}$ [26,35,37,55] when the $Ca^{2+}$ binding affinity of the EF-hand region of both isoforms is identical [72]. Our data showed that the inactivation by divalent ions arises from the same substructures and proceeds by the same mechanism in both RyR isoforms, similar to the activation. In both activation and inactivation, the differences between isoforms emerge from the differences in the strength of the binding signal transmission to the channel gate. According to the COI model, in the case of activation, this influences the allosteric coefficient $f_{Ca}$ that drives RyR opening, which is lower in RyR2 than in RyR1, and thus causes a higher maximum open probability of RyR2. In the case of inactivation, the difference is manifested in the lower allosteric coefficient $h_M$ in RyR1 than in RyR2, which leads to a higher sensitivity to inactivation in RyR1 than in RyR2.

The limited data on RyR2 single-channel activity in the presence of ATP suggest that this ligand increases the transmission of the allosteric signal from the activation site, resulting in two effects: An increase of the apparent calcium sensitivity and maximum open probability, evoked by the decrease of the allosteric coefficient $f_{Ca}$, and an increase of the sensitivity to inactivation (decreased $h_{CaM}$) caused by the interaction between the $Ca^{2+}$-bound activation site and the $M^{2+}$-bound inhibition site. The increased sensitivity of RyR2 to $Ca^{2+}$-dependent inactivation in the presence of ATP is also supported by single-channel experiments [26,41]. RyR channels can be considered mostly in the primed state under these conditions since the binding of ATP analogs induces the primed structural macrostate in RyRs even in the absence of $Ca^{2+}$ [73]. Fortunately, the two sets of conditions for RyR2 open probability data that were available in the literature turned out to represent the activation of channels either selectively from the closed state (Fig 10C), or selectively from the primed state (Fig 10A and 10B). The primed state is structurally very close to the open state, and therefore the transitions between them should be energetically less demanding than the transitions between the structurally more different closed and open states. Precise determination of these effects would require expanding the COI model to explicitly include the ATP binding site and the primed state; construction of such a model is at present hampered by the lack of the RyR channel open probability data at a sufficiently wide range of experimental conditions and the absence of high-resolution structures of WT RyR2 in the primed state. Since the COI model does not contain explicit ATP-binding

transitions, the effect of ATP transpired only through the allosteric parameters $f_{Ca}$ for Ca²⁺-dependent activation and $h_{CaM}$ for M²⁺-dependent inactivation.

The ATP site-containing branches of the activation pathway (A1 – A6, Table 6) appear responsible for the synergy between the Ca²⁺ and ATP activating mechanisms in all RyR channels, as previously recognized for the primed and open RyR1 structures [58]. Our analysis reveals that this interaction path also exists in the closed RyR structures 7k0t, 7vmm, and 7ua5 with the activation binding site unoccupied by Ca²⁺, in which the relative position of the C-terminal and Central domains differs from the primed and open channels. This indicates that the molecular backbone for the transmission of the activation allosteric signal is preexistent in all RyR gating states and that the Ca²⁺ binding itself may constitute the signal that promotes the channel activation.

## Limitations

The determination of allosteric pathways was performed on models derived from cryo-EM structures with suboptimal resolution. Especially the regions of the EF-hand loops were not well resolved, and multiple residues of the structure were either represented by alanines or did not satisfactorily fit into the electron density map. Since the topology of the unresolved DR1 region is not fully understood, this region was replaced by a short polyglycine segment. The substantial editing and energy minimization resulted in model structures that contained a negligible amount of clashes at the expense of less ideal geometry but without a significant change in the overall MolProbity scores (see S3 File). The accuracy of the disclosed allosteric networks cannot be exactly determined; therefore, molecular dynamics simulations would be necessary to evaluate their stability.

The single-channel experiments, used here for deriving the model of RyR operation, were performed under a limited number of experimental conditions that also slightly differed among laboratories. Only experiments under two sets of conditions close to the physiological ones were included in the analysis since RyR activity is sensitive to the ion composition and ionic strength of its environment [74–76]. Experiments with purified RyR channels were neither included for their different sensitivity to activation and inactivation by Ca²⁺ [77].

Due to the stochastic nature of the single-channel activity and the limited amount of available data, the precision of some data used in this study might be influenced by natural variance, especially at low open probability conditions. However, sufficiently exact measurements under conditions of low channel activity would require the collection of a large amount of data [78] that might not be experimentally feasible. Therefore, several parameters of the model of RyR operation had to be fixed in this study at previously estimated values [3,30]. To determine the relative importance of the intra- and inter-monomeric inhibition pathways would require quantitative data on the effect of Mg²⁺ on RyR1 and RyR2 channels in the absence of ATP.

Unfortunately, it was not possible to use experimental data on ³H-ryanodine binding for tuning the COI model of RyR operation. First, the data from many laboratories do not present RyR inactivation, which might be caused by the lengthy experimental procedure or by the presence of higher salt concentrations [79]. Second, at least in some laboratories, activation of ³H-ryanodine binding by Mg²⁺ ions was observed [80], an effect that has not been observed in the single-channel measurements. Third, ³H-ryanodine binding data are mostly presented relative to the maximum observed binding, without providing the ³H-ryanodine binding capacity $B_{max}$ that would correspond to the maximum of the channel open probability ($P_o = 1$).

At present, two problems of RyR inactivation have not been solved yet: Regarding the cardiac isoform, no structure of the inhibited RyR2 was published till the present time. Despite observations of over 50% inhibition by Ca²⁺ in the purified RyR2 channel at 10 mM Ca²⁺ in the absence of ATP [81], or even at 1 mM Ca²⁺ in the native RyR2 channel in the presence of ATP [41], the only RyR2 structure published at 5 mM Ca²⁺ in the presence of ATP and caffeine represented the open and not the inactivated state [10]. Regarding the skeletal muscle isoform, no electron densities attributable to ions bound at the EF-hand loops were observed either in the Ca²⁺-inactivated RyR1 [22] or the Mg²⁺-inactivated RyR1 [23]. However,

the local resolution of the electron density map is insufficient to decide unequivocally whether $Ca^{2+}$ or $Mg^{2+}$ ions are present or not in the structure of the apparent ion binding inhibition site.

## Concluding remarks

The existence of the comprehensive model of mammalian RyR operation puts together the partial observations on RyR structures, equilibrium binding studies, and single-channel gating activity, collected across species and over the years by many independent laboratories. This approach may find use in deciphering the allosteric mechanisms of multimeric proteins such as ion channels or receptors in general.

The methods of bioinformatic analysis applied to numerous RyR structures have shown that in both RyR isoforms, the inhibitory binding site for divalent ions is very likely formed by the EF1 and EF2 loops. The transmission of the inhibition signal to the channel gate proceeds through two pathways: the inter-monomeric pathway that leads through the interface between the EF-hand region and the S23* loop, and the intra-monomeric pathway that converges with the $Ca^{2+}$-activation pathway. The difference between RyR1 and RyR2 inhibition by divalent ions is caused by the different coupling efficiency of the inhibition pathways in RyR1 and RyR2. Modulation of the inactivation pathways may be a prospective target for pharmacological interventions in relevant diseases of skeletal and cardiac muscles.

RyR1 and RyR2 structures present a peculiar construction of their core that controls the RyR channel activity in dependence on the cytosolic concentration of $Ca^{2+}$ and $Mg^{2+}$ ions and endogenous modulators like ATP and xanthines. Each RyR monomer has specific binding sites for these ligands connected to the channel gate by a common allosteric network. Changes in the ligand occupation of binding sites drive stochastic transitions of RyR monomers among the closed – primed – open – inactivated macrostates towards the energetically more preferable state. This means that RyR can assume any macrostate with some probability for a time under any experimental conditions. The macrostate of monomers in the RyR homo-tetramer does change in synchrony, by a structural process supported by the inter-monomeric coupling of allosteric networks. Recent data point to the central role of the U-motif at which the intra-monomeric and inter-monomeric allosteric networks merge and thus allow for the exchange of energy.

## Methods

### Bioinformatics analyses

Ryanodine receptor sequences were taken from the UniProt database (https://www.uniprot.org/) and aligned using CLUSTAL Omega at EMBL-EBI (https://www.ebi.ac.uk/jdispatcher/msa/clustalo; [82]) relative to rabbit RyR1. The residue numbers of the examined domains are given in Table 9 for the studied species/isoforms.

Sequence identities of RyR isoforms were calculated using the SIAS server (http://imed.med.ucm.es/Tools/sias.html). In the text, sequence numbers are given for the rabbit RyR1 (rRyR1) and human RyR2 (hRyR2), if not indicated otherwise.

Structures of ryanodine receptors were obtained from the protein data bank (https://www.rcsb.org/). To ensure compatibility of structural and functional data, we have considered and used those RyR structures that were either ligand-free or contained $Ca^{2+-}$, $Mg^{2+}$, caffeine (xanthine), ATP (ACP) visible in their structure or were obtained in the presence of these ligands or their combination (S1 File). These included 26 structures of rabbit RyR1 [5,7,22,71], 7 structures of pig RyR2 [10,83], 5 structures of human RyR2 [8] and 5 structures of mouse RyR2 [66]. The resolution of these structures was reported in the range of 2.45 - 6.2 Å. Structures of mutant RyRs, or structures obtained in the presence of ryanodine, ruthenium red, or the activator pentachlorobiphenyl, and structures with unresolved critical amino acid side chains were not included.

The flexion angle between the horizontal plane and the line formed by Cα atoms of residues corresponding to rRyR1 residues 348 and 1052 [24] was measured manually in Chimera Ver. 1.17 [84] (https://www.rbvi.ucsf.edu/chimera) and is shown in the S1 File.

Editing and optimization of structures were performed in Chimera (Ver. 1.17) as follows: The monomer core (the C-terminal quarter residues 3639 – 5037; [85]) or its selected part was cut out from the whole RyR structure. In model structures used for the determination of allosteric pathways, the short unresolved loops were reconstructed to correspond to the known sequence, and the missing long part of the unresolved DR1 region was replaced by a polyglycine loop, using the MODELLER [86] add-on to Chimera. The reconstructed loops were optimized using MODELLER [86] or the server MODLOOP (https://modbase.compbio.ucsf.edu/modloop/ [87,88]). The edited structures were repaired by replacing incomplete residues using the Dunbrack library [89]. Finally, hydrogens and appropriate charges were added. Selective residue mutations were introduced to the studied segments of interest in Chimera.

The repaired model structures were energy-minimized using at least 1000 iterations of steepest descent and 100 iterations of maximum gradient minimization with fixed positions of the first five and last five Cα atoms. The resulting model structures were used for different types of analysis as indicated in S1 File.

The maximum distance between Cα atoms of H-bond-forming electron donor residues glutamate or aspartate and the electron acceptor residues lysine or arginine was calculated as follows: The distance from the Cα atom to the farthest oxygen atom of the electron donor residue ($d_D$), and the distance from the Cα atom to the farthest nitrogen atom of the electron acceptor residue ($d_A$) in their extended configurations were measured. The maximum distance was calculated as the sum ($d_D + d_A + 2.7$) Å, where 2.7 Å is the standard length of a hydrogen bond.

The EF-hand identity score (IS) was calculated according to the scoring system of Zhou et al. [90], which compares the identity of amino acids in the ion-binding loop of the examined EF-hand sequence with the 12 amino acids of the canonical EF-hand loop (EF-hand PROSITE pattern PS00018) or their allowed equivalents (D, N, S at position 1, and any residue at position 2). The resulting identity score consists of three numbers x/y/z. The value of x (0–12) reports the number of amino acids concurring at their given positions with amino acids of the allowed EF-hand loop residues. The value of y (0–6) reports the number of amino acids concurring with the 6 amino acids of the loop in positions 1, 3, 5, 7, 9, and 12, involved in divalent ion binding. The value of z (0–3) compares the amino acid concurrence at positions 1, 3, and 12, considered critical for ion binding. Thus, a canonical EF hand, such as that of calmodulin, provides the highest EF-hand identity score of 12/6/3. We report the IS together with the sum of the partial scores in the form of x/y/z (x+y+z).

The ion binding score (IBS) and the number of ion binding poses (NIBP) of EF-hand loops were estimated on edited and energy-optimized model structures using the MIB2 server http://combio.life.nctu.edu.tw/MIB2/ [51] as follows: first, the EF-hand region and 31 subsequent residues ending with a β-hairpin, corresponding to rabbit RyR1 residues 4032–4182, were isolated from the selected RyR structures (S1 File). This segment was chosen for its expected structural stability. The quality of the optimized structures as determined by the MolProbity server [91] is reported in the S3 File. The optimized EF-hand structures were uploaded to the MIB2 server, which returned the ion-binding scores for all individual residues and the number of acceptable ion-binding poses.

Electrostatic interactions as well as the interface area between the interacting EF-hand region and S23 loop were quantified on edited and optimized structures using the server PDBePISA (https://www.ebi.ac.uk/msd-srv/prot_int/cgi-bin/piserver). The examined structures contained residues 4032–4182 of the extended EF-hand region and residues 4695–4745 of the S23* loop.

Root-mean-square deviations (RMSD) were estimated on unedited original structures in Chimera (Ver. 1. 17). The examined structures were aligned through the Cα atoms of their Central domain residues (corresponding to rRyR1 residues 3668–3679, 3694–3733, 3753–3855, and 3873–4070, which were resolved in all RyR structures) with the corresponding Cα atoms of the reference RyR structures 7k0t (closed RyR1) and 7tdg (inactivated RyR1). The RMSD between Cα atoms of the examined RyR domains in different structural states and the corresponding Cα atoms of the reference RyR structures were calculated. RMSD calculation of the S23 loop was performed only on residues 4716–4754 positioned in the vicinity of the neighbor monomer since the rest of the S23 loop was not well-resolved in all structures. Otherwise, the domain residues were specified according to Table 9. The RMSD values were normalized for 100 amino acid residues

**Table 9. Sequence numbers of the ryanodine receptor core domains.**

| Domain | rRyR1 | mRyR2 | hRyR2 | pRyR2, rRyR2 |
|---|---|---|---|---|
| Central domain (CD) | 3668-4070 | 3634-4024 | 3635-4025 | 3636-4026 |
| EF-hand region (EF) | 4071-4131 | 4025-4085 | 4026-4086 | 4027-4087 |
| U-motif (U) | 4132-4251 | 4086-4205 | 4087-4206 | 4088-4207 |
| S23 loop (S23) | 4663-4786 | 4591-4715 | 4592-4716 | 4593-4717 |
| S45 linker (S45) | 4821-4834 | 4750-4763 | 4751-4764 | 4752-4765 |
| S6 segment (S6) | 4910-4956 | 4839-4885 | 4840-4886 | 4841-4887 |
| C-terminal domain (CTD) | 4957-5033 | 4886-4962 | 4887-4963 | 4888-4964 |

Prefixes r, m, h, and p refer to rabbit, mouse, human, and pig.

[56] to account for differences in the length of individual domains and denoted as 7k0t-RMSD$_{100}$ or 7tdg-RMSD$_{100}$ for alignment with 7k0t or 7tdg, respectively. Only RMSD$_{100}$ values larger than 2 Å were considered to indicate a significant structural change [92].

Allosteric pathways were traced using the server OHM (https://dokhlab.med.psu.edu/ohm/#/home, [57]), in which the allosteric pathway is determined on the basis of the network of contacts between pairs of residues in the given structure. Five RyR1 and four RyR2 structures with the best resolution and the lowest number of unresolved residues were selected from the studied set (S1 File) and processed for analysis. Two copies of the monomer core were aligned to two adjacent monomers of the original RyR tetrameric structure. The resulting dimers were subjected to energy minimization in Chimera Ver. 1.17 until all clashes were removed. The quality of the models as determined by the MolProbity server [91] is reported in the S3 File. Finally, the optimized model structures were uploaded to the OHM server to trace the sequences of residues connecting the allosteric site of interest with the active site. As the allosteric site, either the residues of the EF1 and EF2 loops (INH) as the putative divalent ion binding inhibition site or the CD and CTD residues of the Ca$^{2+}$-binding activation site (ACT) [57,58] were selected. As the active site, the gate and hinge residues of the S6 segment (Table 4) were selected. The simulation of the perturbation propagation for calculating the residue importance was performed 10,000 times per structure.

The output of the OHM server consisted of two items for each RyR structure: (1) The list of the first 100 most important allosteric pathways ("_paths.txt"), in which a chain of interacting residues represented the individual pathway. This list was used for sorting paths by important residues and estimating the fraction of major pathways and branches. (2) The description of the complete allosteric network ("_net.txt"), consisting of the list of critical residues of the allosteric network with their respective residue importance (RI), and the list of network edges, showing connections between the residues in the pathway. The network list was used to select the residues with the highest RI and examine their connections within the pathways. In addition to these outputs, the graph of the allosteric network for a selected set of pathways (3, 5, 7, 10, 20, 30, 40, 50, 60, 70, 80, 90, or all of the 100 listed pathways) was visually examined to understand the meaning of different pathways in the context of all possible pathways. The fractional contribution of critical residues (output list "_net.txt") functionally and structurally important to the allosteric network was calculated as the frequency of their occurrence in the first 100 most frequently occurring pathways (output list "_paths.txt"). The output files "_net.txt" and "_paths.txt" are located in S2 File.

## Modeling the RyR operation

The model of RyR operation was constructed on the Monod-Wyman-Changeaux (MWC) theorem [48] and homo-tetrameric Markovian memory-free mechanism [3,59] using a statistical mechanics approach [30,49]. The model presumes an allosteric mechanism where ions activating and/or inhibiting RyR opening bind to the specific regulatory sites present on each monomer and exert their effect by allosteric coupling to the remote gating site. The gating site of the RyR pore should be in the

open position in all four monomers to make the ion permeation possible. The free energy of ion binding to the specific binding site is supposed to be independent of the identity of the monomer and the presence/absence of ligands at the binding sites of other monomers but is supposed to depend on the macrostate of the whole RyR. Mathematically, all possible macrostates pre-exist in the absence of ligands and the binding of a ligand affects the relative occurrence of a given macrostate by altering its free energy. A transition between macrostates occurs in a concerted manner at all monomers [93]. This supposition greatly reduces the number of considered macrostates and reduces demands on computational power.

The dependence of the RyR open probability on the concentration of regulatory ions was described using the framework of statistical mechanics [30,49]. The parameters of the model equations (see Results) were optimized using the Levenberg-Marquardt method for the best fit of the RyR single-channel open probability data in OriginPro (ver. 2022b). The simple shape of the calcium dependence of open probability and the limited size of available experimental data led to the mutual dependence of some fitted parameters and non-convergence of the fit. To solve this problem, the number of free parameters was reduced as described in the Results. A global fit of the whole dataset was then performed with the minimal chi-square value as the minimization criterion for the whole dataset.

## Statistical analysis

One-way ANOVA was performed in Origin (OriginLab, V. 2024) to test the significance of the differences between groups. The test was performed only if the distribution of data did not significantly deviate from normality (Shapiro-Wilk test), and if the population variances were not significantly different (Levene's test). Comparison of means was performed with Scheffe's test. Values with $p < 0.05$ were considered significant. The power of the test was calculated at $\alpha = 0.05$ and had to be $> 0.95$ to accept the ANOVA results. If the distribution of data deviated from normality, the Kruskal-Wallis ANOVA followed by Dunn's test was used instead.

## Supporting information

**S1 Fig. A detailed view of the EF-hand region and the S23 loop taken from the open, primed and inactivated RyR structures aligned at their S23 loops.** The different orientation of the EF-hand relative to the S23 loop in RyR1 vs. RyR2 is indicated by the red double arrows. Note the similar orientation of the EF-hand region relative to the S23 loop in all structures of the same isoform. This figure complements Fig 5.
(TIF)

**S2 Fig. Examples of the allosteric networks in RyR1 and RyR2.** Allosteric and activation sites: purple. Activation pathways: red. Inhibition pathways: green. Common activation and inhibition pathways: black. ATP/ACP molecule: yellow. This figure complements Fig 8.
(TIF)

**S3 Fig. Transitions between and within macrostates of the RyR tetramer.** The overall representation of the model cannot be depicted on a two-dimensional plane, therefore the transitions between microstates are shown only for the binding of the first ion ($Ca^{2+}$ or $Mg^{2+}$) to the activation site and the first divalent ion $M^{2+}$ to the inhibition site. Other states and transitions can be constructed in the same way. The number of $Ca^{2+}$ ions bound to the activation site is i = 0–4; the number of $Mg^{2+}$ ions bound to the activation site is j = 0–4-i, and the number of $M^{2+}$ ions bound to the inhibition site is k = 0 – 4. Squares: closed state; quatrefoils with a central opening: open state; angular quatrefoils: inactivated state. The channels with 1 $Ca^{2+}$ bound to the activation site are shown in red; channels with 1 $Mg^{2+}$ bound to the activation site are shown in pink; channels with 1 $M^{2+}$ bound to the inhibition site are shown in green. Channels with both the activation and the inhibition site occupied by one ion are shown in the red/green or pink/green gradient. The three digits inside the shapes denote respectively, from left to right, the number of $Ca^{2+}$ ions at the activation site, the number of $Mg^{2+}$ ions at the activation site, and the number of $M^{2+}$ ions at the inhibition site. Note that the ion-binding dissociation constants are 4x

larger than those in Fig 9 since in the ion-free tetramer, there are 4 possibilities to bind an ion to each of the binding sites, and in a tetramer with a single bound ion, there is one possibility to dissociate the ion. In addition, since all transitions are reversible, the product of each reaction cycle in the clockwise and anticlockwise directions is equal. Transitions between macrostates are shown only for the ligand-free channel but are present between all same-colored channels. This figure complements Fig 9.
(TIF)

**S1 Table. Fractional occurrence of branches in the inhibition network pathways.** This table complements Table 9.
(PDF)

**S2 Table. Fractional occurrence of branches in the activation network pathways.** This table complements Table 6.
(PDF)

**S3 Table. Interaction between the activation network and inhibition network pathways.** This table complements Table 7.
(PDF)

**S4 Table. Structured methods - reagents and tools table.**
(DOCX)

**S1 Movie. The allosteric pathways in the inactivated RyR1 structure 7tdg.** Allosteric and activation sites: purple. Activation pathway: red. Inhibition pathways: green. Common activation and inhibition pathway: black. ACP molecule: yellow.
(WMV)

**S2 Movie. The allosteric pathways in the open RyR2 structure 7ua9.** Allosteric and activation sites: purple. Activation pathway: red. Inhibition pathways: green. Common activation and inhibition pathway: black. ATP molecule: yellow.
(WMV)

**S1 File. List of analyzed structures and their properties.** The Excel table contains the following data: PDB accession code; species; experimental conditions; reported resolution (Å); RyR isoform; assigned structural state; $Ca^{2+}$ concentration (μM); the presence of ATP (1) or ACP (2); the presence of caffeine (1) or xanthine (2); the presence of FKBP; the presence of CaM; use of the original PDB for distance analysis; use of the original PDB for RMSD analysis; use of the optimized extended EF-hand region for MIB2 analysis; use of the scrambled optimized extended EF-hand region for MIB2 analysis; use of the optimized extended EF-hand region and S23 loop for the PDBePISA analysis; use of the mutated optimized extended EF-hand region and S23 loop for the PDBePISA analysis; use of the optimized dimer for OHM analysis; pore diameter at the Cα atoms of I4937 (Å); pore diameter at the Cα atoms of N4933 (Å); flexion angle; distance D1 between the Cα atoms of E4075.A and R4736.D (Å); distance D2 between the Cα atoms of K4101.A and D4730.D (Å).
(XLSX)

**S2 File. Description of the input and output files of the OHM server.** The zipped dataset contains the structural description of the input files as well as all data output of the OHM server for the 9 analyzed structures. The content of individual directories and files of the S2 File is described in the file "Description of files.txt".
(ZIP)

**S3 File. Stereochemical accuracy and geometry of the structures and models used in the study.** The Excel file contains the results of the analysis by the MolProbity server. The output of the server is provided for three stages of model optimization: structures with the atom coordinates identical to the experimental structure (Original), structures after adding all missing atoms in partially resolved residues but lacking the missing regions (Completed), and structures with

completed missing regions and minimized model energy (Optimized). "Segments for MIB2 analysis" – structures used for the analysis of divalent ion binding to the EF-hand region (20 structures). "Core dimers for OHM analysis" – structures used for the analysis of allosteric pathways by the OHM server (9 structures).
(XLSX)

**S4 File. Source data.** The Excel file contains the X and Y values of the data points in Figs 2, 3, 4, and 10.
(XLSX)

## Acknowledgments

The authors would like to thank Dr. N. V. Dokholyan and Dr. V.R. Chirasani for their help with using the OHM server, and to Dr. B. Iaparov for the help with using statistical mechanics.

## Author contributions

**Conceptualization:** Alexandra Zahradníková, Ivan Zahradník.

**Data curation:** Alexandra Zahradníková.

**Formal analysis:** Alexandra Zahradníková, Jana Pavelková, Miroslav Sabo.

**Funding acquisition:** Alexandra Zahradníková, Sefer Baday.

**Investigation:** Alexandra Zahradníková, Sefer Baday, Ivan Zahradník.

**Methodology:** Alexandra Zahradníková, Ivan Zahradník.

**Project administration:** Alexandra Zahradníková.

**Supervision:** Alexandra Zahradníková.

**Validation:** Alexandra Zahradníková, Ivan Zahradník.

**Visualization:** Alexandra Zahradníková, Ivan Zahradník.

**Writing – original draft:** Alexandra Zahradníková, Ivan Zahradník.

**Writing – review & editing:** Alexandra Zahradníková, Jana Pavelková, Miroslav Sabo, Sefer Baday, Ivan Zahradník.

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
