## [Decision Letter · Decision Letter 0]

22 Jan 2025

PCOMPBIOL-D-24-02096

Structure-based mechanism of RyR channel operation by calcium and magnesium ions

PLOS Computational Biology

Dear Dr. Zahradnikova,

Thank you for submitting your manuscript to PLOS Computational Biology. After careful consideration, we feel that it has merit but does not fully meet PLOS Computational Biology's publication criteria as it currently stands. Therefore, we invite you to submit a revised version of the manuscript that addresses the points raised during the review process.

Please submit your revised manuscript within 30 days Mar 24 2025 11:59PM. If you will need more time than this to complete your revisions, please reply to this message or contact the journal office at ploscompbiol@plos.org. Please include the following items when submitting your revised manuscript:

We look forward to receiving your revised manuscript.

Kind regards,

Arne Elofsson

Section Editor

PLOS Computational Biology

Arne Elofsson

Section Editor

PLOS Computational Biology

**Journal Requirements:**

1) Please provide an Author Summary. This should appear in your manuscript between the Abstract (if applicable) and the Introduction, and should be 150-200 words long. The aim should be to make your findings accessible to a wide audience that includes both scientists and non-scientists. Sample summaries can be found on our website under Submission Guidelines:

3) "Redline version of the manuscript" is currently uploaded as an 'Supporting Information' file type; please upload it as 'Manuscript with Highlighted changes".

4) Please ensure that the funders and grant numbers match between the Financial Disclosure field and the Funding Information tab in your submission form. Note that the funders must be provided in the same order in both places as well. State the initials, alongside each funding source, of each author to receive each grant. For example: "This work was supported by the National Institutes of Health (####### to AM; ###### to CJ) and the National Science Foundation (###### to AM).".

**Reviewers' comments:**

Reviewer's Responses to Questions

**Comments to the Authors:**

Reviewer #1: This is a technical paper on an important topic for the RyR channel and ion channel communities, but it is of more limited interest to the general public as no direct applications can be gleaned from it.

Reviewer #2: The authors addressed satisfactorily my questions in my previous review. Based on the improvements and the scientific advance that this work represents, I recommend publication.

**Have the authors made all data and (if applicable) computational code underlying the findings in their manuscript fully available?**

Reviewer #1: **No: ** no evidence has been provided on the geometry or stereochemical accuracy (standard vs non-standard rotamers, Ramachandran plots, clash score and other metrics) of the residues in the structures/sub-structures that have been used in the above-mentioned calculations and subsequent analysis.

Reviewer #2: Yes

PLOS authors have the option to publish the peer review history of their article (what does this mean? ). If published, this will include your full peer review and any attached files.

**Do you want your identity to be public for this peer review?** For information about this choice, including consent withdrawal, please see our Privacy Policy .

Reviewer #1: No

Reviewer #2: No

**Figure resubmission:**
---

## [Editor Report · Decision Letter 1]

11 Mar 2025

Dear Dr. Zahradnikova,

We are pleased to inform you that your manuscript 'Structure-based mechanism of RyR channel operation by calcium and magnesium ions' has been provisionally accepted for publication in PLOS Computational Biology.

Best regards,

Arne Elofsson

Section Editor

PLOS Computational Biology

Arne Elofsson

Section Editor

PLOS Computational Biology

---

## [Editor Report · Acceptance letter]

PCOMPBIOL-D-24-02096R1

Structure-based mechanism of RyR channel operation by calcium and magnesium ions

Dear Dr Zahradníková,

I am pleased to inform you that your manuscript has been formally accepted for publication in PLOS Computational Biology. Your manuscript is now with our production department and you will be notified of the publication date in due course.

With kind regards,

Lilla Horvath
